# Dorsal raphe nucleus to anterior cingulate cortex 5-HTergic neural circuit modulates consolation and sociability

Laifu Li[1,2†], Li-Zi Zhang[1†], Zhi-Xiong He[1], Huan Ma[1], Yu-Ting Zhang[1], Yu-Feng Xun[1], Wei Yuan[1,3], Wen-Juan Hou[1], Yi-Tong Li[1], Zi-Jian Lv[1], Rui Jia[1], Fa-Dao Tai[1]*

[1]Institute of Brain and Behavioral Sciences, College of Life Sciences, Shaanxi Normal University, Xi'an, China; [2]College of Life Sciences, Nanyang Normal University, Nanyang, China; [3]Provincial Key Laboratory of Acupuncture and Medications, Shaanxi University of Chinese Medicine, Xianyang, China

**Abstract** Consolation is a common response to the distress of others in humans and some social animals, but the neural mechanisms underlying this behavior are not well characterized. By using socially monogamous mandarin voles, we found that optogenetic or chemogenetic inhibition of 5-HTergic neurons in the dorsal raphe nucleus (DR) or optogenetic inhibition of serotonin (5-HT) terminals in the anterior cingulate cortex (ACC) significantly decreased allogrooming time in the consolation test and reduced sociability in the three-chamber test. The release of 5-HT within the ACC and the activity of DR neurons were significantly increased during allogrooming, sniffing, and social approaching. Finally, we found that the activation of 5-HT1A receptors in the ACC was sufficient to reverse consolation and sociability deficits induced by the chemogenetic inhibition of 5-HTergic neurons in the DR. Our study provided the first direct evidence that DR-ACC 5-HTergic neural circuit is implicated in consolation-like behaviors and sociability.

*For correspondence:
taifadao@snnu.edu.cn

†These authors contributed equally to this work

## Introduction

Consolation behavior, which is referred to as an increase in affiliative contact toward a distressed individual by an uninvolved bystander, is an important component of the social capabilities of humans (*de Waal and Preston, 2017*; *Field et al., 2009*). Impaired consolation has been frequently observed in many psychiatric diseases, such as depression, autism, and schizophrenia (*Young et al., 2015*). According to de Waal's multilevel conceptualization of empathy (*de Waal, 2008*; *de Waal and Preston, 2017*), consolation represents an intermediate level of empathy (the primary level of 'emotional contagion', the more complex level of 'consolation', and the most elaborate level of 'perspective taking and targeted helping'), which has long been assumed to exist in species possessing complex cognitive functions, such as humans, apes, dolphins, and elephants (*Pérez-Manrique and Gomila, 2018*); however, recent studies have indicated that it also exists in some socially lived rodents, such as prairie voles (*Burkett et al., 2016*), mandarin voles (*Li et al., 2019*), and rats (*Knapska et al., 2010*).

Currently, studies of the neural mechanisms underlying consolation and other forms of empathy have primarily focused on the oxytocin systems (*Burkett et al., 2016*; *Li et al., 2019*). However, as a complex social behavior, consolation may require the coordinated actions of numerous neuromodulators and neurotransmitters. Serotonin (5-HT) is an evolutionarily ancient neurotransmitter that has long been implicated in a variety of emotional disorders (*Faye et al., 2020*; *Garcia-Garcia et al., 2018*; *Meneses and Liy-Salmeron, 2012*). According to recent studies, 5-HT transmission is also involved in a series of social behaviors such as social interaction (*Walsh et al., 2018*), social reward,

and aggression (*Dölen et al., 2013*). Regarding empathy, a recent study revealed an association between salivary 5-HT levels and the empathic abilities of people (*Matsunaga et al., 2017*); a polymorphism in the promoter region of the 5-HT transporter gene has been linked to individual differences in empathy (*Gyurak et al., 2013*); and MDMA (±3,4 methylenedioxymethamphetamine, better known as the recreational drug 'ecstasy'), which is well known to stimulate a feeling of closeness and empathy in its users (*Carlyle et al., 2019*), has been confirmed to robustly increase the release of 5-HT in an activity-independent manner (*Heifets and Malenka, 2016*). In animal studies, *Kim et al., 2014* found that microinjection of 5-HT into the anterior cingulate cortex (ACC) impairs vicarious fear and alters the regularity of neural oscillations in mice. Our recent study has indicated that 5-HT1A receptors within the ACC are involved in consolation deficits induced by chronic social defeat stress in mandarin voles (*Li et al., 2020*). However, to our knowledge, direct evidence for an association between 5-HT and consolation has yet to be obtained.

Dorsal raphe nucleus (DR) is a main source of 5-HT neurons and provides 70% of 5-HTergic projections in the forebrain (*Fu et al., 2010*; *Luo et al., 2015*). DR 5-HTergic neurons form dense, broad, and bidirectional neural connections with a broad range of forebrain and limbic structures, including the ACC (*Celada et al., 2013*; *Charnay and Léger, 2010*), which is a central hub for various types of empathy. Therefore, direct modulation of the DR→ACC 5-HTergic circuit to investigate its functional role in consolation-like behaviors is interesting and meaningful.

The released 5-HT binds to pre- and postsynaptic receptors. To date, at least 14 different 5-HT receptor subtypes have been identified in the brain (*Artigas, 2013*). Among them, 5HT1AR and 5HT2AR are the two main subtypes that are expressed at high levels in the prefrontal cortex (*Carhart-Harris and Nutt, 2017*; *Santana and Artigas, 2017*). The distribution, signaling pathways, and functions of these two receptors are substantially different, and both receptors play critical roles in modulating cortical activity and neural oscillations (*Celada et al., 2013*). Previous studies have indicated that 5HT2AR gene single nucleotide polymorphisms are associated with empathy-related social communication abilities (*Gong et al., 2015*), and a 5HT2AR agonist increases emotional empathic ability (*Dolder et al., 2017*). However, in animal studies by *Kim et al., 2014*, blockade of 5-HT receptors in the ACC did not affect observational fear responses in mice. Clearly, the specific functions of 5HT1AR and 5HT2AR in empathy-like behaviors still require further examination.

The mandarin vole (*Microtus mandarinus*) is a socially monogamous rodent that is widely distributed across China (*He et al., 2019*). As shown in our previous studies, this species is capable of displaying consolation-like behaviors upon exposure to a distressed partner (*Li et al., 2020*; *Li et al., 2019*). In the present study, we first investigated the function of DR→ACC 5-HTergic circuits in consolation-like behaviors using optogenetic and chemogenetic approaches. To provide more direct evidence, we then monitored ACC 5-HT release and DR neuron activities during this behavior by using in vivo fiber photometry. Finally, we used chemogenetics plus pharmacological approaches to investigate which types of 5-HT receptors in the ACC are involved in consolation-like behaviors in mandarin voles. In order to investigate any potential sex differences during these processes, both male and female subjects were included in our study. As consolation is, in general, a pro-social behavior, some social behaviors were investigated during the tests.

## Results

### Optogenetic inhibition of DR 5-HT neurons in the DR→ACC neural circuit impaired consolation and reduced sociability

We first determined the 5-HTergic projection relationship between the DR and ACC in mandarin voles. For this experiment, the retrograde tracer cholera toxin subunit B (CTB) was injected into the ACC, followed by immunofluorescence staining of the DR sections with tryptophan hydroxylase 2 (TPH2), a marker of 5-HTergic neurons. A substantial number of TPH2+ neurons colocalized with CTB, indicating the presence of DR-ACC 5-HTergic projections (*Figure 1—figure supplement 1*). We then used a novel dual-virus optogenetics approach to explore the function of the DR-ACC 5-HTergic circuit in consolation-like behaviors and sociability, where double-floxed AAV-DIO-ChR2-mCherry (DIO-ChR2) or AAV-DIO-eNpHR3.0-mCherry (DIO-eNpHR3.0) was injected into the DR and retro-AAVs containing the TPH2 promoter and Cre element (rAAV(Retro)-TPH2-Cre) were injected

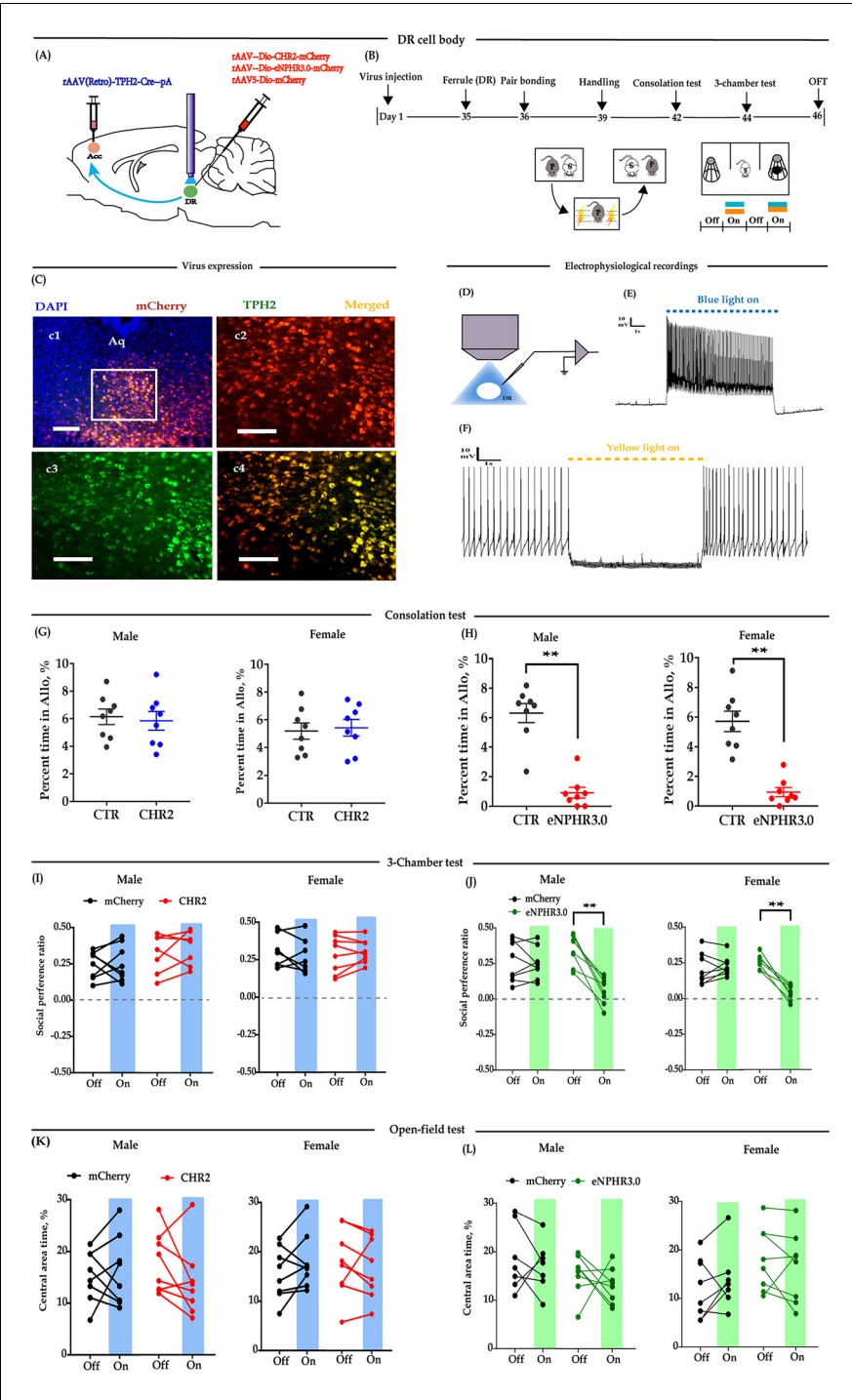

**Figure 1.** Optogenetic bidirectional modulation of 5-HT neuron in the DR in the DR-ACC neural circuit. (**A**) Schematic of optogenetic manipulation. (**B**) Timeline of experiments. (**C**) Immunohistological image showing virus expression in the DR (c1) and amplified images in the left box showing the mCherry, TPH2, and the colocalization of the two ('c2-c4'). (**D**) Electrophysiological recording model. (**E and F**) Representative traces from electrophysiological recordings showing photostimulation (**E**) and photoinhibition of a 5-HT neuron (**F**). (**G**) Quantification of allogrooming time in the consolation test of the CHR2 and control animals ($n = 8$ in each group; CHR2 vs CTR, independent samples $t$-test and Bayesian independent samples $t$-test; male: $t_{(14)} = 0.340$, p=0.739, $BF_{+0} = 0.445$ with median posterior $\delta = 0.104$, 95% CI = [−0.673 to 0.930]; female: $t_{(14)} = -0.279$, p = 0.785, $BF_{+0} = 0.439$ with median posterior $\delta = -0.085$, 95% CI = [−0.906 to 0.694]). (**H**) Quantification of allogrooming time in the consolation test of the eNPHR3.0 and control animals ($n = 8$ in each group; CHR2 vs eNPHR3.0, independent

*Figure 1 continued on next page*

*Figure 1 continued*

samples *t*-test and Bayesian independent samples *t*-test; male: $t_{(14)}$ = 7.293, p < 0.001, $BF_{+0}$ = 6000.583; female: $t_{(14)}$ = 6.327, p < 0.001, $BF_{+0}$ = 1562.921). (I) Quantification of social preference ratio in the three-chamber test of the CHR2 and control animals (*n* = 7 in CHR2 groups, one male and one female were excluded from analysis due to immobility; *n* = 8 in CTR groups; two-way repeated measures ANOVA along with two-way Bayesian repeated measures ANOVA; male: group: $F_{(1, 13)}$ = 3.042, p = 0.105, $BF_{(incl)}$ = 1.184; light: $F_{(1, 13)}$ = 0.531, p = 0.479, $BF_{(incl)}$ = 0.425; group × light: $F_{(1, 13)}$ = 0.246, p = 0.628, $BF_{(incl)}$ = 0.479; female: group: $F_{(1, 13)}$ = 2.088, p = 0.172, $BF_{(incl)}$ = 0.790; light: $F_{(1, 13)}$ = 0.180, p = 0.678, $BF_{(incl)}$ = 0.426; group × light: $F_{(1, 13)}$ = 0.233, p = 0.638, $BF_{(incl)}$ = 0.358). (J) Quantification of social preference ratio in the three-chamber test of the eNPHR3.0 and control animals (*n* = 8 in each group; two-way repeated measures ANOVA along with two-way Bayesian repeated measures ANOVA; male: group: $F_{(1, 14)}$ = 4.084, p = 0.063, $BF_{(incl)}$ = 1.236; light: $F_{(1, 14)}$ = 28.361, p < 0.001, $BF_{(incl)}$ = 25.390; group × light: $F_{(1, 14)}$ = 22.959, p < 0.001, $BF_{(incl)}$ = 87.850; post-hoc comparisons (Tukey): mCherry_Off vs mCherry_On, p = 0.981; eNPHR_Off vs eNPHR_On, p<0.001; female: group: $F_{(1, 14)}$ = 11.892, p = 0.004, $BF_{(incl)}$ = 4.965; light: $F_{(1, 14)}$ = 22.678, p < 0.001, $BF_{(incl)}$ = 7.067; group × light: $F_{(1, 14)}$ = 33.771, p < 0.001, $BF_{(incl)}$ = 623.339; post-hoc comparisons (Tukey): mCherry_Off vs mCherry_On, p = 0.879; eNPHR_Off vs eNPHR_On, p < 0.001). (K) Quantification of time spent in the central area in the open-field test of the CHR2 and control animals (*n* = 8 in each group; two-way repeated measures ANOVA along with two-way Bayesian repeated measures ANOVA; male: group: $F_{(1, 14)}$ = 0.009, p = 0.465, $BF_{(incl)}$ = 1.184; light: $F_{(1, 14)}$ = 0.808, p = 0.384, $BF_{(incl)}$ = 0.442; group × light: $F_{(1, 14)}$ = 2.266, p = 0.155, $BF_{(incl)}$ = 0.964; female: group: $F_{(1, 14)}$ = 0.240, p = 0.632, $BF_{(incl)}$ = 0.602; light: $F_{(1, 14)}$ = 0.341, p = 0.568, $BF_{(incl)}$ = 0.371; group × light: $F_{(1, 14)}$ = 2.192, p = 0.161, $BF_{(incl)}$ = 0.910). (L) Quantification of time spent in the central area in the open-field test of the eNPHR3.0 and control animals (male_mCherry, *n* = 7 (one was excluded from analysis due to immobility); male_eNPHR3.0, *n* = 8; female_mCherry, *n* = 8; male_eNPHR3.0, *n* = 8; two-way repeated measures ANOVA along with two-way Bayesian repeated measures ANOVA; male: group: $F_{(1, 13)}$ = 6.326, p = 0.026, $BF_{(incl)}$=1.935; light: $F_{(1, 13)}$ = 1.176, p = 0.298, $BF_{(incl)}$=0.605; group × light: $F_{(1, 13)}$ = 0.039, p = 0.846, $BF_{(incl)}$ = 0.578; post-hoc comparisons (Tukey): mCherry_Off vs mCherry_On, p = 0.928; eNPHR_Off vs eNPHR_On, p = 0.785; female: group: $F_{(1, 14)}$ = 0.794, p = 0.388, $BF_{(incl)}$ = 0.660; light: $F_{(1, 14)}$ = 0.632, p = 0.440, $BF_{(incl)}$ = 0.402; group × light: $F_{(1, 14)}$ = 3.390, p = 0.087, $BF_{(incl)}$ = 1.352; post-hoc comparisons (Tukey): mCherry_Off vs mCherry_On, p = 0.928; eNPHR_Off vs eNPHR_On, p = 0.785). Scale bars, 100 μm. Error bars are ± SEM. **p < 0.01. For raw data in this figure, please refer to *Figure 1—source data 1*. ACC: anterior cingulate cortex; Aq: aqueduct; ANOVA: analysis of variance; CTR: control; DR: dorsal raphe nucleus; TPH2: tryptophan hydroxylase 2; 5-HT: serotonin.

The online version of this article includes the following source data and figure supplement(s) for figure 1:

**Source data 1.** Source data indicating behavioral performances of bidirectional optogenetic modulation of 5-HT neurons in the DR in the DR-ACC neural circuit.

**Figure supplement 1.** The histology of CTB injecting into the right ACC of male (upper row) and female voles (lower row).

**Figure supplement 2.** Immunohistological images showing colocalization of opsins (mCherry, red), TPH2 + neurons (green), and DAPI (blue) in the DR of male (A) and (B) female voles.

**Figure supplement 3.** Schematics depicting virus spread (shades) and optic fiber placements (lines) for recording and functional manipulation experiments, related to *Figures 1*, *3*, *4* and *5* (A, B, C, and D, respectively).

**Figure supplement 4.** Effect of bidirectional optogenetic modulation of DR 5-HT neuron activities in the DR-ACC neural circuit on some control behaviors.

**Figure supplement 4—source data 1.** Source data indicating bidirectional optogenetic modulation of DR 5-HT neuron activities in the DR-ACC neural circuit on some control behaviors.

**Figure supplement 5.** Effect of optogenetic inhibition of DR 5-HT neurons in the DR-ACC neural circuit does not elicit long-lasting effects (< 24 hr) on allogrooming and chasing behavior in the consolation test.

**Figure supplement 5—source data 1.** Source data indicating the optogenetic inhibition of DR 5-HT neurons in the DR-ACC neural circuit does not elicit long-lasting effects.

into the ACC (*Figure 1A*). This virus strategy ensures that opsins (excitatory CHR2 or inhibitory eNpHR3.0) are mainly expressed within the DR-ACC 5-HTergic circuit. Immunohistochemical staining showed that more than 75% of mCherry-labeled neurons expressed TPH2 in both male and female voles, and more than 80% of TPH2+ cells coexpressed mCherry (*Figure 1—figure supplement 2*). In the electrophysiological study, we found that the DR neurons reliably responded to pulses of 473 (activation)/593 (inhibition) nm light stimuli (*Figure 1D–E*). These results indicate the viability of this virus strategy.

To test whether modulation of DR 5-HT neuron activity alters consolation and sociability, 5 weeks after the virus injection, optic fibers were implanted above the DR (*Figure 1A–B*). The virus

expression sites and the optic fiber placement schematics are shown in *Figure 1—figure supplement 3A*. According to Burkett's and our previous results (*Burkett et al., 2016*; *Li et al., 2019*), the time spent grooming their stressed partners (allogrooming) is an important indicator of consolation-like behaviors. We then exposed the subjects to their electric-shocked partners and recorded the time spent allogrooming, chasing (closely following), and selfgrooming (consolation test; for details, please refer to 'Materials and methods'). The three-chamber test was used to assay the sociability along with open-field anxiety and locomotion (*Figure 1B*). In the CHR2-expressing animals, there was no conclusive evidence that optogenetic activation of 5-HTergic neurons affected their time spent in allogrooming, chasing, and selfgrooming in the 10-min consolation test (*Figure 1G*, *Figure 1—figure supplement 4A–D*), the social interaction ratio in the three-chamber test (*Figure 1I*), and behavioral performance in the open-field test (*Figure 1K*, *Figure 1—figure supplement 4E*).

However, in the eNPHR3.0-expressing animals, there was extremely strong evidence that optogenetic inhibition of 5-HT neurons in the DR reduced the time spent allogrooming (*Figure 1H*) and chasing (*Figure 1—figure supplement 4B*), but had little effect on the control behavior of selfgrooming (*Figure 1—figure supplement 4D*). Following experiments indicated that the light-inhibition effects significantly reduced within 24 hr (*Figure 1—figure supplement 5*). In the three-chamber test, light inhibition also significantly reduced the social preference ratio (*Figure 1J*). In the open-field test, the results were inconclusive, which suggests that the data for the time spent in the central area and the total distance traveled are equally likely under H0 and H1 (*Figure 1L*, *Figure 1—figure supplement 4F*).

The above results indicate that although we have no conclusive results for optogenetic activation, optogenetic inhibition of 5-HT neurons within the DR→ACC 5-HTergic circuit impaired consolation-like behavior and sociability in mandarin voles.

## Optogenetic inhibition of ACC 5-HT terminals in the DR→ACC neural circuit similarly impaired consolation and reduced sociability

Modulation of 5-HT neurons may affect other neurons in the DR and thus confound behavioral performance. In subsequent experiments, we placed optic fibers in the ACC and investigated whether the direct modulation of 5-HT terminals in this region would exert the same effects (*Figure 2A*). Similarly, we found that optogenetic activation of DR-ACC 5-HT terminals did not significantly alter behavioral performance in the consolation test (*Figure 2C*, *Figure 2—figure supplement 1A and C*) and open-field test (*Figure 2G*, *Figure 2—figure supplement 1E*). In the three-chamber test, two-way repeated measures analysis of variance (ANOVA) revealed a moderate 'light × group' interaction in females ($F_{(1, 14)}$ = 6.297, p = 0.025, $BF_{incl}$ = 4.186). Subsequent post-hoc comparison results showed that light stimulation slightly increased the social preference ratio in CHR2-expressing females but not in mCherry-expressing females (ChR2: off vs on, p = 0.033; mCherry: off vs on, p = 0.975; *Figure 2E*). The 'light × group' interaction was not evidently observed in males ($F_{(1, 14)}$ = 3.556, p = 0.08, $BF_{incl}$ = 1.239).

Optogenetic inhibition of DR-ACC 5-HT terminals significantly reduced the time spent allogrooming and chasing in the consolation test (*Figure 2D*, *Figure 2—figure supplement 1B*) and the social preference ratio in the three-chamber test (*Figure 2F*). Similarly, the inhibition had little effect on the control behavior of selfgrooming (*Figure 2—figure supplement 1D*) and behavioral performance in the open-field test (*Figure 2H*, *Figure 2—figure supplement 1F*), and the inhibitory effect lasted no more than 24 hr (*Figure 2—figure supplement 2*). The above results indicated that optogenetic modulation of ACC 5-HT terminals has nearly identical effects as modulation of DR 5-HT neurons.

## Chemogenetic inhibition of DR 5-HT neurons in the DR-ACC neural circuit impaired consolation and reduced sociability

To confirm the above optogenetic results over a longer time frame, we used a chemogenetic approach to selectively express 'Gq-DREADD' or 'Gi-DREADD' in DR 5-HT neurons by injecting AAV-DIO-hM4Dq-mCherry (Gq-DREADD) or AAV-DIO-hM4Di-mCherry (Gi-DREADD) into the DR and rAAV(Retro)-TPH2-Cre into the ACC (*Figure 3A*). Immunohistochemical staining revealed that more than 65% of TPH2-labeled neurons were infected by mCherry virus and more than 60% of mCherry cells coexpressed TPH2 (*Figure 3—figure supplement 1*). The virus expression site

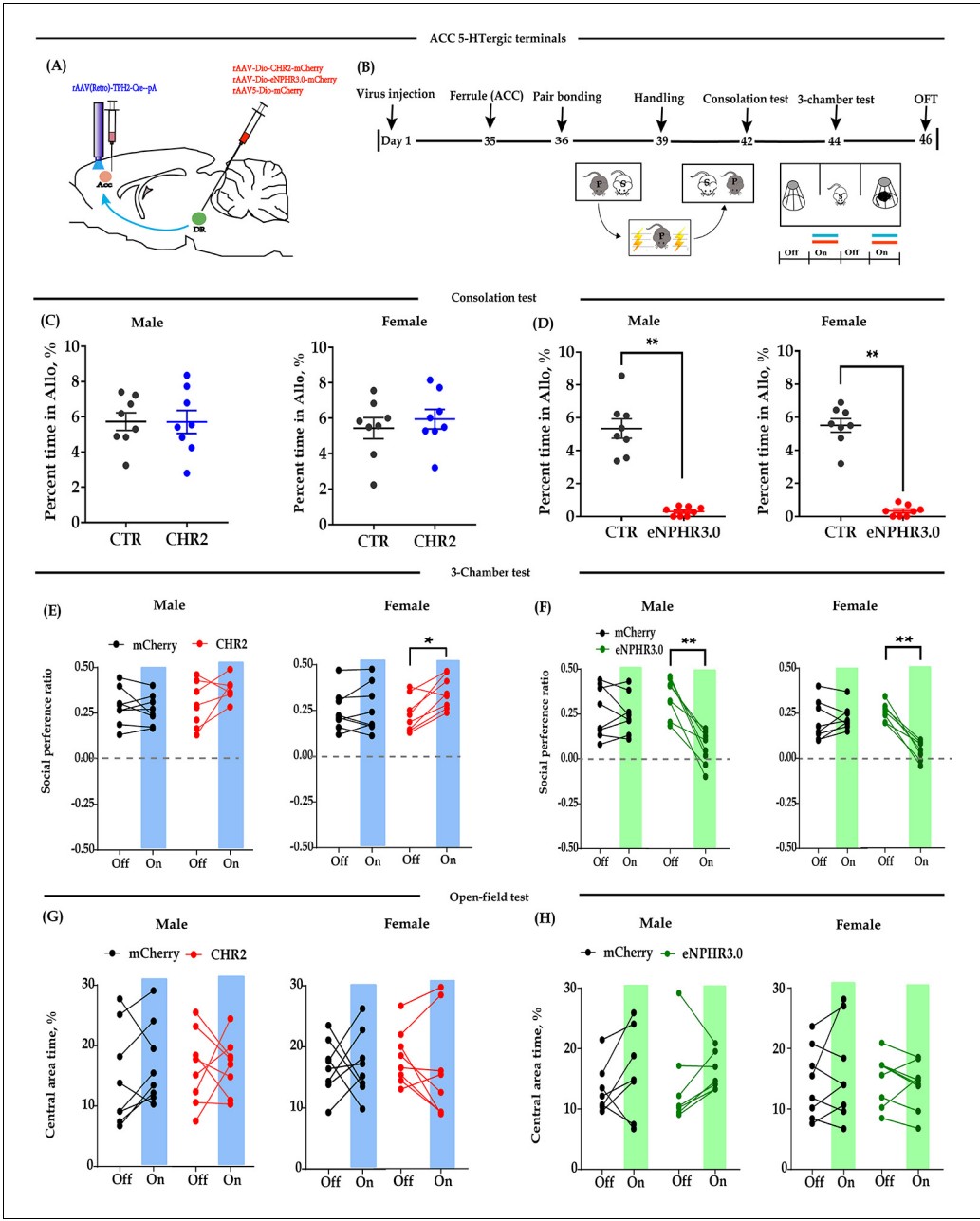

**Figure 2.** Optogenetic bidirectional modulation of 5-HT terminals within the ACC in the DR-ACC neural circuit.
(A) Schematic of optogenetic manipulation. (B) Timeline of experiments. (C) Quantification of time spent in allogrooming in the consolation test of the CHR2 and control animals ($n$ = 8 in each group; CHR2 vs CTR, independent samples $t$-test and Bayesian independent samples $t$-test; male: $t_{(14)}$ = 0.011, p = 0.992, $BF_{+0}$ = 0.428 with median posterior $\delta$ = 0.003, 95% CI = [−0.792 to 0.800]; female: $t_{(14)}$ = −0.630, p = 0.539, $BF_{+0}$ = 0.489 with median posterior $\delta$ = −0.194, 95% CI = [−1.054 to 0.575]). (D) Quantification of time spent in allogrooming in the consolation test of the eNPHR3.0 and control animals ($n$ = 8 in each group; CHR2 vs CTR, independent samples $t$-test and Bayesian independent samples $t$-test; male: $t_{(14)}$ = 8.44, p<0.001, $BF_{+0}$ = 26556.455; female: $t_{(14)}$ = 12.174, p<0.001, $BF_{+0}$ = 1.577 × e$^{+6}$). (E) Quantification of social preference ratio in the three-chamber test of the CHR2 and control animals ($n$ = 8 in each group; two-way repeated measures ANOVA along with two-way Bayesian repeated measures ANOVA; male: group: $F_{(1, 14)}$ = 0.443, p = 0.516, $BF_{(incl)}$ = 0.624; light: $F_{(1, 14)}$ = 5.764, p = 0.031, $BF_{(incl)}$ = 1.763; group × light: $F_{(1, 14)}$ = 3.556, p = 0.080, $BF_{(incl)}$ = 1.239; female: group: $F_{(1, 14)}$ = 3.985, p = 0.066, $BF_{(incl)}$ = 1.151; light: $F_{(1, 14)}$ = 3.704, p = 0.075, $BF_{(incl)}$ = 1.097; group × light: $F_{(1, 14)}$ = 6.297, p = 0.025, $BF_{(incl)}$ = 4.186; post-hoc comparisons (Tukey): mCherry_Off vs mCherry_On, p = 0.975; CHR2_Off vs CHR2_On, p = 0.033). (F) Quantification of social preference ratio in the three-chamber test of the eNPHR3.0 and control

*Figure 2 continued on next page*

*Figure 2 continued*

animals ($n$ = 8 in each group; two-way repeated measures ANOVA along with two-way Bayesian repeated measures ANOVA; male: group: $F_{(1, 14)}$ = 9.763, p = 0.007, $BF_{(incl)}$ = 3.374; light: $F_{(1, 14)}$ = 13.168, p = 0.003, $BF_{(incl)}$=9.588; group × light: $F_{(1, 14)}$ = 10.843, p = 0.005, $BF_{(incl)}$ = 13.892; post-hoc comparisons (Tukey): mCherry_Off vs mCherry_On, p = 0.995; eNPHR_Off vs eNPHR_On, p = 0.001; female: group: $F_{(1, 14)}$ = 6.912, p = 0.020, $BF_{(incl)}$ = 1.545; light: $F_{(1, 14)}$ = 93.330, p < 0.001, $BF_{(incl)}$ = 423.687; group × light: $F_{(1, 14)}$ = 65.062, p < 0.001, $BF_{(incl)}$ = 20440.982; post-hoc comparisons (Tukey): mCherry_Off vs mCherry_On, p = 0.679; eNPHR_Off vs eNPHR_On, p < 0.001). (G) Quantification of time spent in the central area in the open-field test of the CHR2 and control animals ($n$ = 8 in each group; two-way repeated measures ANOVA along with two-way Bayesian repeated measures ANOVA; male: group: $F_{(1, 14)}$ = 4.189 × $e^{-4}$, p = 0.984, $BF_{(incl)}$ = 0.483; light: $F_{(1, 14)}$ = 0.842, p = 0.374, $BF_{(incl)}$ = 0.489; group × light: $F_{(1, 14)}$ = 0.501, p = 0.491, $BF_{(incl)}$ = 0.486; female: group: $F_{(1, 14)}$ = 0.014, p = 0.906, $BF_{(incl)}$ = 0.457; light: $F_{(1, 14)}$ = 0.050, p = 0.826, $BF_{(incl)}$ = 0.335; group × light: $F_{(1, 14)}$ = 0.177, p = 0.681, $BF_{(incl)}$ = 0.440). (H) Quantification of time spent in the central area in the open-field test of the eNPHR3.0 and control animals (male: $n$ = 7, two animals (one from each group) were excluded from analysis due to immobility; female $n$ = 8; two-way repeated measures ANOVA along with two-way Bayesian repeated measures ANOVA; male: group: $F_{(1, 12)}$ = 0.091, p = 0.769, $BF_{(incl)}$ = 0.566; light: $F_{(1, 12)}$ = 8.130, p = 0.015, $BF_{(incl)}$ = 4.162; group × light: $F_{(1, 12)}$ = 0.802, p = 0.388, $BF_{(incl)}$ = 0.550; post-hoc comparisons (Tukey): mCherry_Off vs mCherry_On, p = 0.532; eNPHR_Off vs eNPHR_On, p = 0.086; female: group: $F_{(1, 14)}$ = 0.276, p = 0.607, $BF_{(incl)}$ = 0.578; light: $F_{(1, 14)}$ = 0.580, p = 0.459, $BF_{(incl)}$ = 0.404; group × light: $F_{(1, 14)}$ = 2.234, p = 0.157, $BF_{(incl)}$ = 0.871). Error bars are ± SEM. *p < 0.05, **p < 0.01. For raw data in this figure, please refer to *Figure 2— source data 1*. ACC: anterior cingulate cortex; ANOVA: analysis of variance; DR: dorsal raphe nucleus; CTR: control; 5-HT: serotonin.

The online version of this article includes the following source data and figure supplement(s) for figure 2:

**Source data 1.** Source data indicating behavioral performances of optogenetic bidirectional modulation of 5-HT terminals within the ACC in the DR-ACC neural circuit.

**Figure supplement 1.** Effect of bidirectional optogenetic modulation of ACC 5-HT terminals in the DR-ACC neural circuit on some control behaviors.

**Figure supplement 1—source data 1.** Source data indicating optogenetic bidirectional modulation of ACC 5-HT terminals in the DR-ACC neural circuit on some control behaviors.

**Figure supplement 2.** Effect of optogenetic inhibition of ACC 5-HT terminals in the DR-ACC neural circuit does not elicit long-lasting effects (<24 hr) on allogrooming and chasing behaviors in the consolation test.

**Figure supplement 2—source data 1.** Source data indicating the optogenetic inhibition of ACC 5-HT terminals in the DR-ACC neural circuit does not elicit long-lasting effects.

---

schematics are shown in *Figure 1—figure supplement 3B*. To determine whether the ligand clozapine N-oxide (CNO) can activate or inhibit DR 5-HT neurons, whole-cell current-clamp recordings were performed. The results showed that the addition of 10 µM CNO remarkably increased the number of action potentials in the Gq-DREADD-transfected neurons (*Figure 3D*). In contrast, CNO caused a significant decrease in the number of spikes (*Figure 3E*) and increased the spike rheobase during current step injections in Gi-DREADD-transfected neurons (*Figure 3F*). These results indicate the specificity and viability of this virus strategy.

In subsequent behavioral studies, we found that CNO (1 mg/kg, intraperitoneally (i.p.))-treated Gi-DREADD-expressing voles showed reduced grooming toward their shocked partners in the consolation test (*Figure 3G*) and decreased sociability in the three-chamber test (*Figure 3H*). The CNO-treated males also showed a trend of spending less time in the open-field test, but the statistics were inconclusive (*Figure 3I*; treatment: $F_{(1,19)}$ = 4.305, p = 0.052, $BF_{(incl)}$ = 4.438; group: $F_{(2,19)}$ = 2.064, p = 0.155, $BF_{(incl)}$ = 0.384; treatment × group: $F_{(1,19)}$ = 0.164, p = 0.850, $BF_{(incl)}$ = 0.300). The treatment had no detectable effects on the total distance traveled in the open-field test (*Figure 3— figure supplement 2C*) and some control behaviors in the consolation test (*Figure 3—figure supplement 2A & B*). In Gq-DREADD and virus control subjects, CNO treatment had no detectable effects on the behavioral performance in all the tests (*Figure 3*).

Based on the results of the chemogenetic and optogenetic experiments described above, we concluded that inhibition of DR-ACC 5-HTergic circuit activity was sufficient to impair consolation and sociability in mandarin voles.

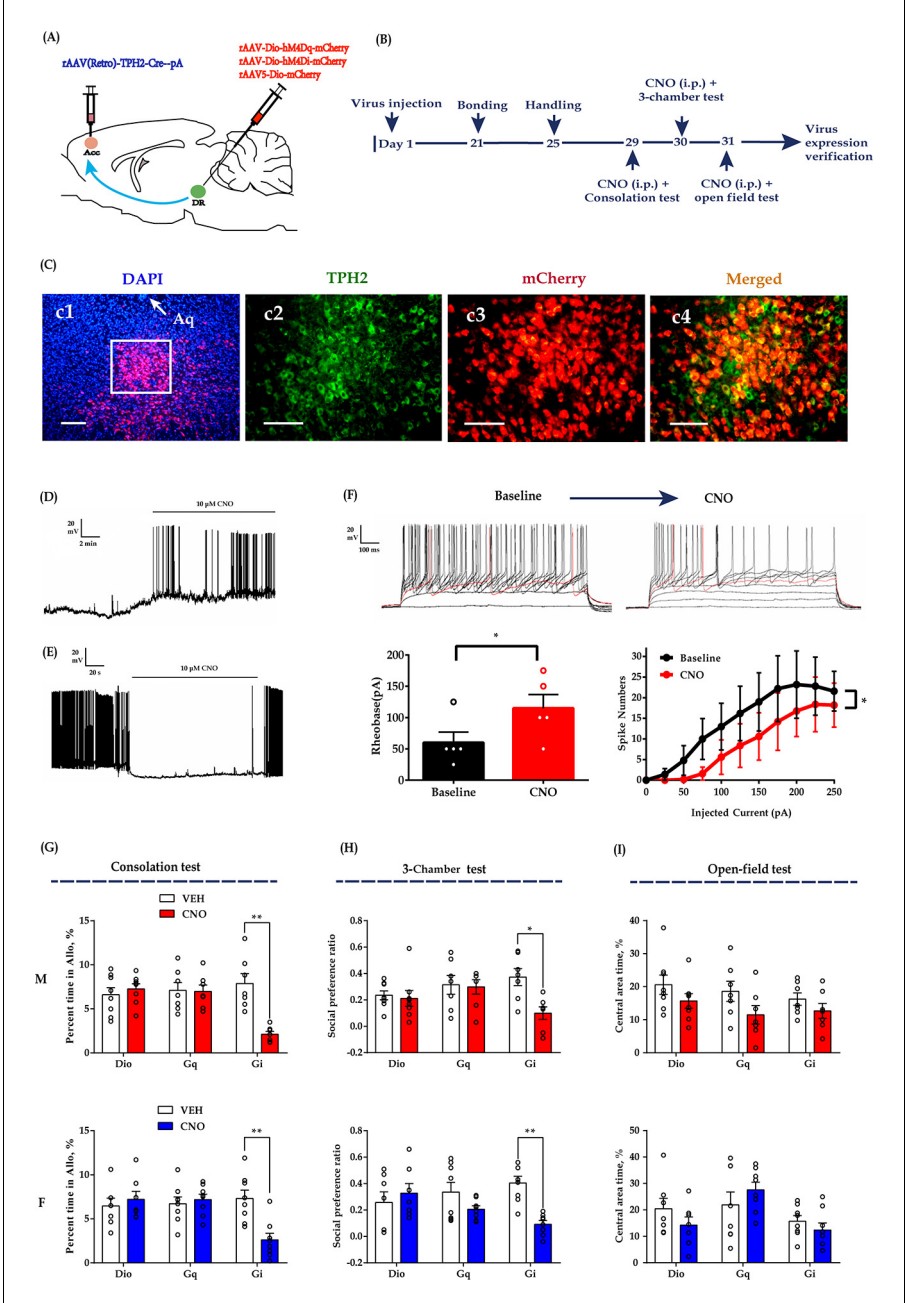

**Figure 3.** Chemogenetic modulation of DR 5-HT neuron activities in the DR-ACC neural circuit. (**A**) Schematic of chemogenetical manipulations. (**B**) Timeline of experiments. (**C**) Immunohistological image showing virus expression in the DR (c1) and amplified images in the left white box showing the mCherry, TPH2, and the colocalization of the two (c2–c4). (**D**) Representative trace from a Gq-DREADD neuron. (**E**) Representative trace from a Gi-DREADD-transfected neuron. (**F**) Quantification of spike rheobase and spike numbers under current step injections in Gi-DREADD-transfected neurons ($n$ = 5 neurons; spike rheobase: paired $t$-test, $t_{(4)}$ = 4.491, p = 0.0109; two-way repeated measures ANOVA; spike numbers: treatment: $F_{(1, 4)}$ = 8.734, p = 0.0417, current: $F_{(10, 40)}$ = 8.989, p<0.0001; treatment ×current: $F_{(10, 40)}$ = 4.013, p = 0.0008). (**G**) Quantification of allogrooming time in the consolation test (two-way repeated measures ANOVA along with two-way Bayesian repeated measures ANOVA; male: treatment: $F_{(1, 19)}$ = 6.300, p = 0.021, $BF_{(incl)}$ = 2.007; group: $F_{(2, 19)}$ = 5.434, p = 0.014, $BF_{(incl)}$ = 0.777; treatment ×group: $F_{(1, 19)}$ = 8.433, p = 0.002, $BF_{(incl)}$ = 215.751; post-hoc comparisons (Tukey): Gi_Saline vs Gi_CNO, p = 0.002; Gq_Saline vs Gq_CNO, p = 1; Dio_Saline vs Dio_CNO, p = 0.991). Female: treatment: $F_{(1, 20)}$ = 4.570, p = 0.045, $BF_{(incl)}$ = 1.120; group: $F_{(2, 20)}$ = 2.884, p = 0.079, $BF_{(incl)}$ = 0.816; treatment × group: $F_{(1, 20)}$ = 10.778, p< 0.001, $BF_{(incl)}$ = 154.421. Post-hoc comparisons (Tukey): Gi_Saline vs Gi_CNO, p < 0.001; Gq_Saline

*Figure 3 continued on next page*

*Figure 3 continued*

vs Gq_CNO, p = 0.996; Dio_Saline vs Dio_CNO, p = 0.977. (**H**) Quantification of social preference ratio in the three-chamber test (two-way repeated measures ANOVA along with two-way Bayesian repeated measures ANOVA; male: treatment: $F_{(1, 19)} = 4.707$, p = 0.043, $BF_{(incl)} = 2.261$; group: $F_{(2, 19)} = 1.387$, p = 0.274, $BF_{(incl)} = 0.365$; treatment × group: $F_{(1, 19)} = 2.990$, p = 0.074, $BF_{(incl)} = 2.562$; post-hoc comparisons (Tukey): Gi_Saline vs Gi_CNO, p = 0.046; Gq_Saline vs Gq_CNO, p = 1; Dio_Saline vs Dio_CNO, p = 1; female: treatment: $F_{(1,20)} = 11.687$, p = 0.003, $BF_{(incl)} = 9.390$; group: $F_{(2, 20)} = 0.205$, p = 0.817, $BF_{(incl)} = 0.272$; treatment × group: $F_{(1, 20)} = 8.923$, p = 0.002, $BF_{(incl)} = 42.856$; post-hoc comparisons (Tukey): Gi_Saline vs Gi_CNO, p < 0.001; Gq_Saline vs Gq_CNO, p = 0.307; Dio_Saline vs Dio_CNO, p = 0.892). (**I**) Quantification of time spent in the central area in the open-field test (two-way repeated measures ANOVA along with two-way Bayesian repeated measures ANOVA; male: treatment: $F_{(1, 19)} = 4.305$, p = 0.052, $BF_{(incl)} = 4.438$; group: $F_{(2, 19)} = 2.064$, p = 0.155, $BF_{(incl)} = 0.384$; treatment × group: $F_{(1, 19)} = 0.164$, p = 0.850, $BF_{(incl)} = 0.300$; post-hoc comparisons (Tukey): Gi_Saline vs Gi_CNO, p = 0.963; Gq_Saline vs Gq_CNO, p = 0.605; Dio_Saline vs Dio_CNO, p = 0.838; female: treatment: $F_{(1, 20)} = 0.288$, p = 0.597, $BF_{(incl)} = 0.313$; group: $F_{(2, 20)} = 4.419$, p = 0.026, $BF_{(incl)} = 2.335$; treatment × group: $F_{(1, 20)} = 2.242$, p = 0.132, $BF_{(incl)} = 1.146$; post-hoc comparisons (Tukey): Gi_Saline vs Gi_CNO, p = 0.959; Gq_Saline vs Gq_CNO, p = 0.726; Dio_Saline vs Dio_CNO, p = 0.711). Male_Dio, n = 8; female_Dio, n = 7; male_Gq, n = 7; female_Gq, n = 8; male_Gi, n = 7; female_Gi, n = 8. Error bars are ± SEM. Scale bars, 100 µm. *p < 0.05, **p < 0.01. For raw data in this figure, please refer to *Figure 3—source data 1*. ACC: anterior cingulate cortex; Aq: aqueduct; ANOVA: analysis of variance; DR: dorsal raphe nucleus; TPH2: tryptophan hydroxylase 2; M: male; F: female; 5-HT: serotonin.

The online version of this article includes the following source data and figure supplement(s) for figure 3:

**Source data 1.** Source data indicating chemogenetic modulation of DR 5-HT neuron activities in the DR-ACC neural circuit.

**Figure supplement 1.** Immunohistological image showing colocalization of DREADD (mCherry, red), TPH2 + neurons (green), and DAPI (blue) in the DR of male (**A**) and (**B**) female voles.

**Figure supplement 2.** Effect of chemogenetic modulation of DR 5-HT neuron activities in the DR-ACC neural circuit on some control behaviors.

**Figure supplement 2—source data 1.** Source data indicating the effects of chemogenetic modulation of DR 5-HT neuron activities in the DR-ACC neural circuit on some control behaviors.

## DR 5-HT neuron activity increased during allogrooming and social approaching

If the DR-ACC 5-HTergic circuit is involved in consolation and sociability, ACC projecting DR 5-HT neurons may change their activity during the corresponding behaviors. To verify this idea, we performed photometric recording in freely moving animals by injecting Cre-inducible AAV-DIO-GCaMp6, a genetically encoded fluorescent $Ca^{2+}$ sensor, into the DR and rAAV(Retro)-TPH2-Cre into the ACC (*Figure 4A*). 10 days later, fibers were implanted above the DR injection sites. Post-hoc histological analysis revealed that more than 80% $GCaMp6^+$ cells overlapped with $TPH2^+$-expressing cells (*Figure 4—figure supplement 1*). The virus expression sites and the optic fiber placement schematics are shown in *Figure 1—figure supplement 3C*.

During the consolation test, we found that the activity of DR 5-HT neurons was reliably increased during allogrooming, social approaching, and sniffing in both sexes (*Figure 4E1–E2, F1–F2 and G1–G2*). When aligning the fluorescence changes to the end of these social bouts, we found that the activity of 5-HT neurons decreased accordingly before withdrawal from allogrooming and approaching (*Figure 4—figure supplement 2A–B*). Withdrawing from sniffing also showed a similar trend but was not statistically significant (*Figure 4—figure supplement 2C*; pre vs post: male: $t_{(5)} = 1.611$, p = 0.084, $BF_{+0} = 1.589$ with median posterior δ = 0.528, 95% CI = [0.041, 1.363]; female: $t_{(5)} = 1.724$, p = 0.073, $BF_{+0} = 1.778$ with median posterior δ = 0.557, 95% CI = [0.046, 1.415]). The increased activity of DR 5-HT neurons did not simply reflect movement initiation as no fluorescence changes were observed when aligning to running and selfgrooming (*Figure 4H1–H2, I1–I2*). Furthermore, no significant changes in fluorescence signals were detected in DR 5-HT neurons of green fluorescent protein (GFP)-expressing control voles during all the behaviors, confirming that the GCaMp6 signals genuinely indicated neuronal activity and did not simply reflect motion artifacts (*Figure 4E3–E4, F3–F4, G3–G4, H3–H4 and I3–I4*).

The above results provide direct evidence that DR 5-HT neurons within the DR-ACC 5-HTergic circuit are recruited in consolation-like behavior and sociability in mandarin voles.

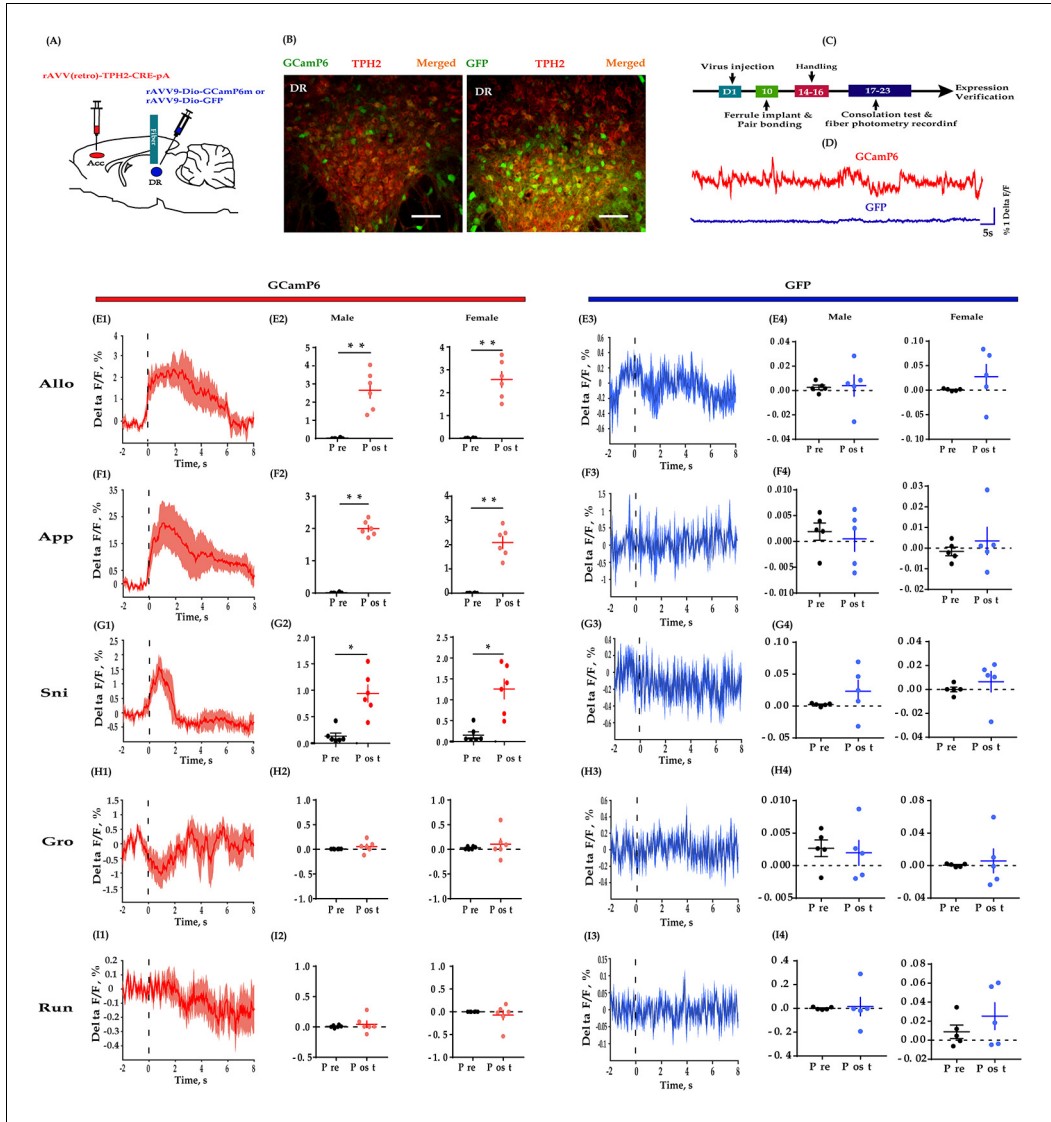

**Figure 4.** Fiber photometry recording DR 5-HT neural dynamics during the consolation test. (**A**) Schematic diagrams depicting the virus injection and recording sites. (**B**) Histology showing the expression of GCaMP6 (left) and GFP control (right) in the DR. (**C**) Experimental timeline for photometry experiments. (**D**) Representative fluorescence changes of GCaMP6 (red line) and GFP (blue line) during photometry recordings. (**E1–I1**) Representative peri-event plot of GCaMP6 fluorescence signals aligned to onsets of various behaviors (for all peri-event plots, the red line denotes the mean signals of four to six bouts of behaviors, whereas the red shaded region denotes the SEM). (**E2**) Quantification of change in GCaMP6 fluorescence signals before and after allogrooming ($n = 6$ in each group; male: $t_{(5)} = -5.967$, $p = 0.002$, $BF_{+0} = 24.488$ with median posterior $\delta = -1.904$, 95% CI = $[-3.749, -0.424]$; female: $t_{(5)} = -7.420$, $p < 0.01$, $BF_{+0} = 52.689$ with median posterior $\delta = -2.397$, 95% CI = $[-4.610, -0.625]$). (**F2**) Quantification of change in GCaMP6 fluorescence signals before and after approaching ($n = 6$ in each group; male: $t_{(5)} = -19.871$, $p < 0.001$, $BF_{+0} = 2233.691$; female: $t_{(5)} = -8.448$, $p < 0.001$, $BF_{+0} = 84.470$ with median posterior $\delta = -2.747$, 95% CI = $[-5.225, -0.767]$). (**G2**) Quantification of change in GCaMP6 fluorescence signals before and after sniffing ($n = 6$ in each group; male: $t_{(5)} = -3.689$, $p = 0.011$, $BF_{+0} = 6.449$ with median posterior $\delta = -1.221$, 95% CI = $[-2.576, -0.138]$; female: $t_{(5)} = -3.689$, $p = 0.014$, $BF_{+0} = 5.312$ with median posterior $\delta = -1.137$, 95% CI = $[-2.434, -0.101]$). (**H2**) Quantification of change in GCaMP6 fluorescence signals before and after selfgrooming ($n = 6$ in each group; male: $t_{(5)} = -1.032$, $p = 0.350$, $BF_{+0} = 0.559$ with median posterior $\delta = -0.300$, 95% CI = $[-1.076, 0.383]$; female: $t_{(5)} = -0.707$, $p = 0.511$, $BF_{+0} = 0.456$ with median posterior $\delta = -0.205$, 95% CI = $[-0.904, 0.466]$). (**I2**) Quantification of change in GCaMP6 fluorescence signals before and after running ($n = 6$ in each group; male: $t_{(5)} = -1.032$, $p = 0.350$, $BF_{+0} = 0.559$ with median posterior $\delta = -0.300$, 95% CI = $[-1.076, 0.383]$; female: $t_{(5)} = -0.707$, $p = 0.511$, $BF_{+0} = 0.456$ with median posterior $\delta = -0.205$, 95% CI = $[-0.904, 0.466]$). (**E3–I3**) Representative peri-event plot of GFP signals aligned to onsets of

*Figure 4 continued on next page*

*Figure 4 continued*

various behavioral events (for all peri-event plots, the blue line denotes the mean signals of four to six bouts of behaviors, whereas the blue shaded region denotes the SE). (E4) Quantification of change in GFP fluorescence signals before and after allogrooming ($n = 5$ in each group; male: $t_{(4)} = -0.145$, $p = 0.885$, $BF_{+0} = 0.401$ with median posterior $\delta = -0.047$, 95% CI = [$-0.3787$, $0.675$]; female: $t_{(4)} = -1.085$, $p = 0.339$, $BF_{+0} = 0.610$ with median posterior $\delta = -0.32$, 95% CI = [$-1.198$,$-0.406$]). (F4) Quantification of change in GFP fluorescence signals before and after approaching ($n = 5$ in each group; male: $t_{(4)} = -1.211$, $p = 0.293$, $BF_{+0} = 0.667$ with median posterior $\delta = -0.371$, 95% CI = [$-1.260$, $0.377$]; female: $t_{(4)} = -0.723$, $p = 0.510$, $BF_{+0} = 0.488$ with median posterior $\delta = -0.220$, 95% CI = [$-1.026$, $0.499$]). (G4) Quantification of change in GFP fluorescence signals before and after sniffing ($n = 5$ in each group; male: $t_{(4)} = 1.001$, $p = 0.373$, $BF_{+0} = 0.577$ with median posterior $\delta = 0.306$, 95% CI = [$-0.427$, $1.156$]; female: $t_{(4)} = -0.687$, $p = 0.530$, $BF_{+0} = 0.479$ with median posterior $\delta = -0.209$, 95% CI = [$-1.009$, $0.510$]). (H4) Quantification of change in GFP fluorescence signals before and after selfgrooming ($n = 5$ in each group; male: $t_{(4)} = 0.237$, $p = 0.825$, $BF_{+0} = 0.407$ with median posterior $\delta = 0.072$, 95% CI = [$-0.647$, $0.819$]; female: $t_{(4)} = -0.350$, $p = 0.744$, $BF_{+0} = 0.418$ with median posterior $\delta = -0.106$, 95% CI = [$-0.865$, $0.610$]). (I4) Quantification of change in GFP fluorescence signals before and after running ($n = 5$ in each group; paired $t$-test and Bayesian paired samples $t$-test, two-tailed; male: $t_{(4)} = -0.202$, $p = 0.850$, $BF_{+0} = 0.404$ with median posterior $\delta = -0.061$, 95% CI = [$-0.806$, $0.659$]; female: $t_{(4)} = -1.813$, $p = 0.144$, $BF_{+0} = 52.689$ with median posterior $\delta = 0.560$, 95% CI = [$-1.580$,$-0.251$]). Error bars are ± SEM. Scale bars, 100 μm. *p<0.05, **p<0.01. Paired samples $t$-test along with Bayesian paired samples $t$-test. For raw data in this figure, please refer to *Figure 4—source data 1*. ACC: anterior cingulate cortex; GFP: green fluorescent protein; TPH2: tryptophan hydroxylase 2; Aq: aqueduct; Allo: allogrooming; Sni: sniffing; App: approaching; Gro: selfgrooming; Run: running; 5-HT: serotonin.
The online version of this article includes the following source data and figure supplement(s) for figure 4:

**Source data 1.** Source data indicating GCaMP6 and GFP fluorescent signals align to some behaviors.
**Figure supplement 1.** Representative viral infection images of GCaMp6 (A) and virus control of GFP (B).
**Figure supplement 2.** GCaMP6 fluorescent signals align to the end of some behaviors.
**Figure supplement 2—source data 1.** Source data indicating GCaMP6 fluorescent signals align to the end of some behaviors.

## ACC 5-HT release increased during allogrooming, social approach, and sniffing

To further establish the potential relevance of 5-HT in consolation-like behaviors, we directly detected the dynamics of 5-HT within the ACC using a newly developed GPCR-activation-based 5-HT sensor along with photometric recording (*Wan et al., 2021*). To do this, we injected AAV encoding the fluorescent 5-HT sensor (AAV-5HTsensor 2.1) unilaterally into the ACC and implanted optic fibers above the injection site 10 days later (*Figure 5A*). The virus expression site and the optic fiber placement schematics are shown in *Figure 1—figure supplement 3D*. Control animals were injected with AAV encoding enhanced GFP (eGFP). Post-hoc histological verification indicated expression of both the 5-HT sensor and eGFP within the ACC (*Figure 5B*).

Next, we measured the dynamics of endogenous 5-HT during the consolation test. Consistent with the above fiber photometry of $Ca^{2+}$ signals in DR 5-HT neurons, we found reliably increased fluctuations in fluorescence within the ACC in conjunction with allogrooming, social approaching, and sniffing, but not with selfgrooming and running (*Figure 5*, left columns). Similarly, the release of 5-HT significantly dropped before the end of allogrooming, approaching, and sniffing (*Figure 5—figure supplement 1*). In control animals that expressed GFP in the ACC, we observed little fluorescence change during any of these behaviors (*Figure 5E3–E4, F3–F4, G3–G4, H3–H4 and I3–I4*).

To validate the selectivity of this 5-HT sensor in mandarin voles, in another set of animals expressing the 5-HT sensor, a 5-HT receptor antagonist metergoline (Met) was injected (i.p., 4 mg/kg) before conducting the fiber photometry experiments. We found that treating voles with Met completely blocked the fluorescence changes of the 5-HT sensor during allogrooming (*Figure 5—figure supplement 2*), validating the viability and specificity of this 5-HT sensors.

The above results provide further evidence that 5-HT released within the ACC may play an important role in consolation-like behaviors and sociability in mandarin voles.

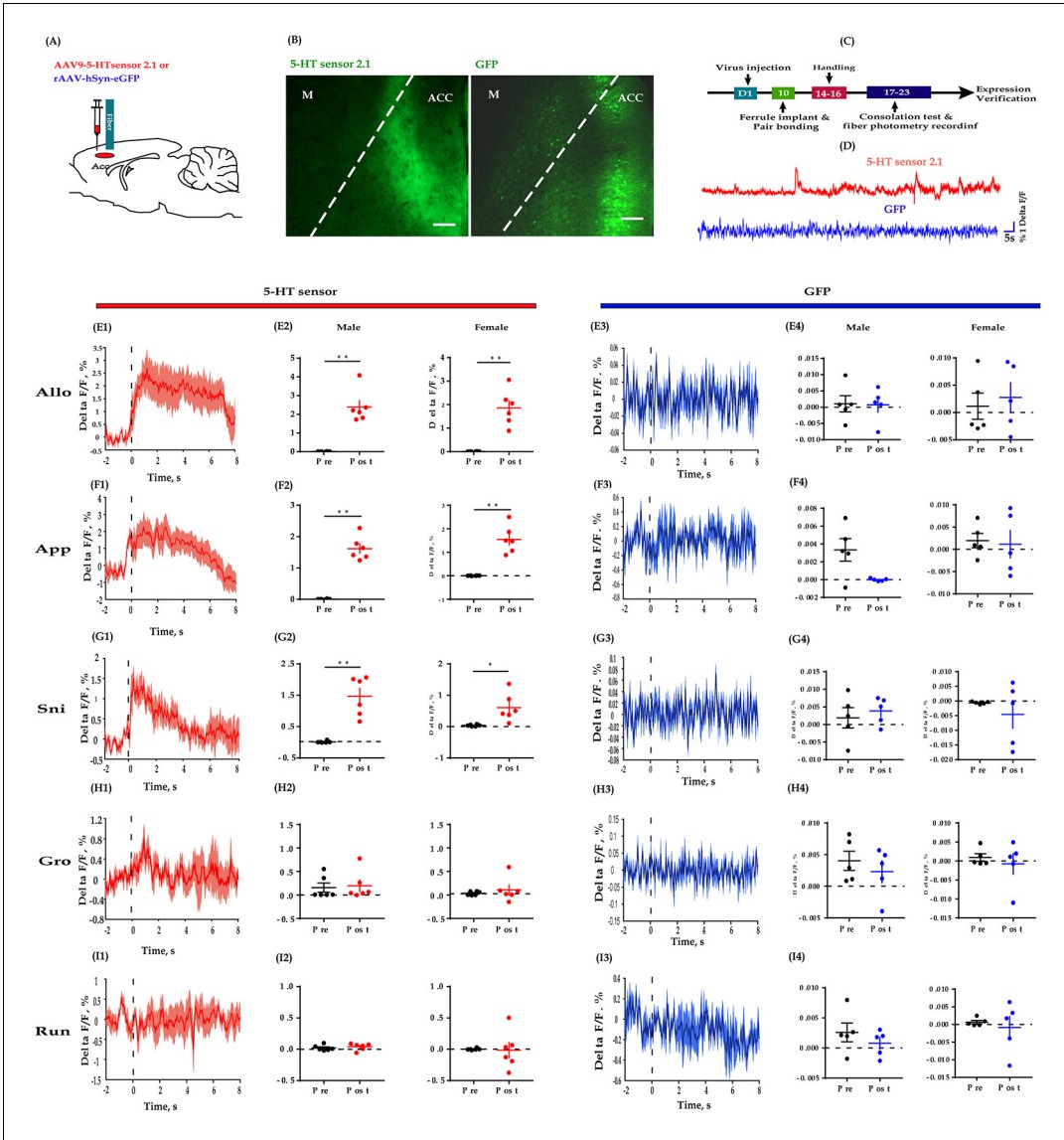

**Figure 5.** Fiber photometry recording dynamics of 5-HT within the ACC during the consolation test. (**A**) Schematic diagrams depicting the virus injection and recording sites. (**B**) Histology showing the expression of 5-HT sensor (left) and GFP control (right) within the ACC. (**C**) Experimental timeline for photometry experiments. (**D**) Representative fluorescence changes of 5-HT sensor (red line) and GFP (blue line) during photometry recordings. (**E1–I1**) Representative peri-event plot of 5-HT fluorescence signals aligned to onsets of various behaviors (for all peri-event plots, the red line denotes the mean signals of four to six bouts of behaviors, whereas the red shaded region denotes the SEM). (**E2**) Quantification of change in 5-HT fluorescence signals before and after allogrooming ($n$ = 6 in each group; male: $t_{(5)}$ = −6.687, p = 0.001, $BF_{+0}$ = 36.393 with median posterior $\delta$ = −2.148, 95% CI = [−4.174,–0.523]; female: $t_{(5)}$ = −6.038, p = 0.002, $BF_{+0}$ = 25.495 with median posterior $\delta$ = −1.928, 95% CI = [−3.790,–0.433]). (**F2**) Quantification of change in 5-HT fluorescence signals before and after approaching ($n$ = 6 in each group; male: $t_{(5)}$ = −10.551, p < 0.001, $BF_{+0}$ = 193.396 with median posterior $\delta$ = −3.460, 95% CI = [−6.490,–1.060]; female: $t_{(5)}$ = −6.496, p = 0.001, $BF_{+0}$ = 32.865 with median posterior $\delta$ = −2.083, 95% CI = [−4.061,–0.497]). (**G2**) Quantification of change in 5-HT fluorescence signals before and after sniffing ($n$ = 6 in each group; male: $t_{(5)}$ = −5.863, p = 0.002, $BF_{+0}$ = 23.053 with median posterior $\delta$ = −1.868, 95% CI = [−3.687,–0.409]; female: $t_{(5)}$ = −3.030, p = 0.029, $BF_{+0}$ = 3.108 with median posterior $\delta$ = −0.921, 95% CI = [−2.070, 0.000]). (**H2**) Quantification of change in 5-HT fluorescence signals before and after selfgrooming ($n$ = 6 in each group; male: $t_{(5)}$ = −0.215, p = 0.838, $BF_{+0}$ = 0.381 with median posterior $\delta$ = −0.062, 95% CI = [−0.753, 0.609]; female: $t_{(5)}$ = −0.738, p = 0.494, $BF_{+0}$ = 0.464 with median posterior $\delta$ = −0.214, 95% CI = [−0.953, 0.458]). (**I2**) Quantification of change in 5-HT fluorescence signals before and after running ($n$ = 6 in each group; male: $t_{(5)}$ = −0.627, p = 0.558, $BF_{+0}$ = 0.438 with median posterior $\delta$ = −0.182, 95% CI = [−0.908,–0.488]; female: $t_{(5)}$ = 0.129, p = 0.903, $BF_{+0}$ =

*Figure 5 continued on next page*

*Figure 5 continued*

0.376 with median posterior $\delta$ = 0.037, 95% CI = [−0.636, 0.722]). (**E3–I3**) Representative peri-event plot of GFP signals aligned to onsets of various behavioral events (for all peri-event plots, the blue line denotes the mean signals of four to six bouts of behaviors, whereas the blue shaded region denotes the SE). (**E4**) Quantification of change in GFP fluorescence signals before and after allogrooming ($n$ = 5 in each group; male: $t_{(4)}$ = 0.082, p = 0.939, $BF_{+0}$ = 0.399 with median posterior $\delta$ = 0.025, 95% CI = [−0.700, 0.760]; female: $t_{(4)}$ = −0.340, p = 0.751, $BF_{+0}$ = 0.610 with median posterior $\delta$ = −0.103, 95% CI = [−0.861,−614]). (**F4**) Quantification of change in GFP fluorescence signals before and after approaching ($n$ = 5 in each group; male: $t_{(4)}$ = −0.836, p = 0.450, $BF_{+0}$ = 0.520 with median posterior $\delta$ = −0.255, 95% CI = [−1.078, 0.469]; female: $t_{(4)}$ = 0.861, p = 0.438, $BF_{+0}$ = 0.528 with median posterior $\delta$ = 0.263, 95% CI = [−0.462, 1.090]). (**G4**) Quantification of change in GFP fluorescence signals before and after sniffing ($n$ = 5 in each group; male: $t_{(4)}$ = 2.686, p = 0.055, $BF_{+0}$ = 2.077 with median posterior $\delta$ = 0.843, 95% CI = [−0.104, 2.086]; female: $t_{(4)}$ = 0.208, p = 0.845, $BF_{+0}$ = 0.405 with median posterior $\delta$ = 0.063, 95% CI = [−0.657, 0.808]). (**H4**) Quantification of change in GFP fluorescence signals before and after selfgrooming ($n$ = 5 in each group; male: $t_{(4)}$ = 0.822, p = 0.457, $BF_{+0}$ = 0.516 with median posterior $\delta$ = 0.251, 95% CI = [−0.473, 1.071]; female: $t_{(4)}$ = 0.653, p = 0.549, $BF_{+0}$ = 0.471 with median posterior $\delta$ = 0.199, 95% CI = [−0.519, 0.994]). (**I4**) Quantification of change in GFP fluorescence signals before and after running ($n$ = 5 in each group; male: $t_{(4)}$ = 1.415, p = 0.230, $BF_{+0}$ = 0.777 with median posterior $\delta$ = 0.435, 95% CI = [−0.331, 1.366]; female: $t_{(4)}$ = 0.510, p = 0.637, $BF_{+0}$ = 0.442 with median posterior $\delta$ = 0.155, 95% CI = [−0.561, 0.932]). Error bars are ± SEM. Scale bars, 100 μm. *p < 0.05, **p < 0.01; paired samples $t$-test along with Bayesian paired samples $t$-test. For raw data in this figure, please refer to *Figure 5—source data 1*. ACC: anterior cingulate cortex; M: motor cortex; TPH2: tryptophan hydroxylase 2; Allo: allogrooming; Sni: sniffing; App: approaching; Gro: selfgrooming; Run: running; GFP: green fluorescent protein; 5-HT: serotonin.

The online version of this article includes the following source data and figure supplement(s) for figure 5:

**Source data 1.** Source data indicating 5-HT sensor and GFP fluorescent signals align to some behaviors.
**Figure supplement 1.** 5-HT sensor fluorescent signals align to the end of some behaviors.
**Figure supplement 1—source data 1.** Source data indicating 5-HT sensor fluorescent signals align to the end of some behaviors.
**Figure supplement 2.** 5-HT sensor fluorescence changes during allogrooming could be blocked by Met.
**Figure supplement 2—source data 1.** Source data indicating 5-HT sensor fluoresce changes during allogrooming could be blocked by Met.

## 5-HT1A receptors are involved in the reduced consolation and sociability induced by chemogenetic inhibition of DR 5-HT neurons

The most abundant 5-HT receptors expressed in the medial prefrontal cortex (mPFC) are 5-HT1AR and 5-HT2AR (*Carhart-Harris and Nutt, 2017*). In our previous study, we indicated that ACC administration of a 5HT1AR agonist (8-OH-DPAT) rescued the impaired consolation and sociability induced by chronic social defeat (*Li et al., 2020*). In the same vein, we used this drug to investigate whether it could relieve the impaired consolation and sociability induced by the inhibition of DR-ACC 5-HTergic circuit, which aimed to clarify the involvement of 5-HT1AR during this process. Considering 5-HT1AR and 5-HT2AR usually exert opposite effects within the mPFC (*Celada et al., 2013*; *Puig and Gulledge, 2011*), and 5-HT2A antagonists had been reported to exert anxiolytic effects in preclinical studies (*Carson and Kitagawa, 2004*), we used MDL 100907 (a 5-HT2A antagonist) here to investigate the possible involvement of 5-HT2AR during this process. To do this, 8-OH-DPAT (200 nl/side; 1.5 mg/ml) or MDL 100907 (200 nl/side; 1 mg/ml) was directly infused into the ACC along with chemogenetic inhibition of DR 5-HT neurons before conducting behavioral assays (*Figure 6A and B*). Not surprisingly, CNO elicited deficits in allogrooming and sociability (*Figure 6C,D,F,G*, left two bars). Pretreatment with 8-OH-DPAT significantly reduced the impact of CNO (*Figure 6C,D*, right two bars; see also the comparison results of 'Vehicle + Saline - Vehicle + CNO' vs '8-OH-DPAT + Saline - 8-OH-DPAT + CNO'; *Figure 6—figure supplement 1A,B*), whereas MDL 100907 had no such effect (*Figure 6F Figure 6G*; right two bars; *Figure 6—figure supplement 1C,D*). Neither the drugs nor viruses had any detectable effect on the control behavior of selfgrooming or behavioral performance in the open-field test (*Figure 6E–H*, *Figure 6—figure supplement 2*).

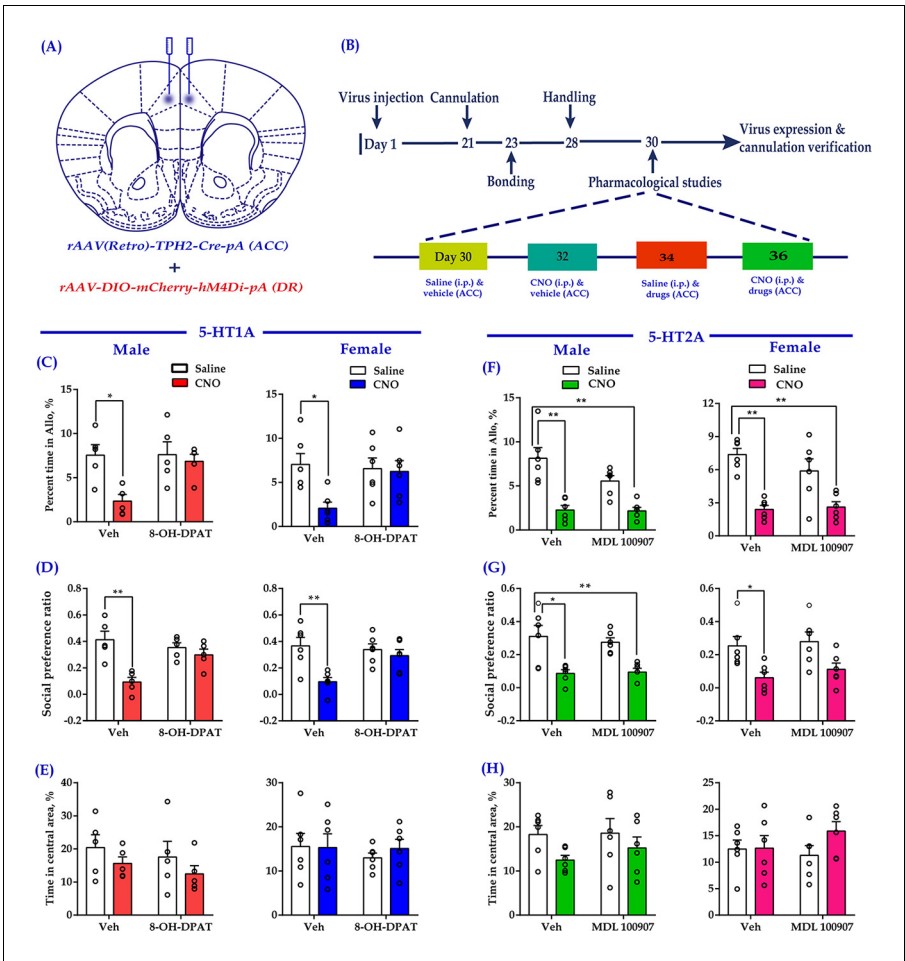

**Figure 6.** Intra-ACC injection of 8-OH-DPAT reduced sociability deficits induced by chemogenetic inhibition of DR 5-HT neurons in the DR→ACC neural circuit. (A) Schematic representation of ACC infusion sites and virus strategy. (B) Timeline of experimental design. (C) Effect of a 5-HT1AR agonist 8-OH-DPAT on allogrooming time in the consolation test ($n = 5$ for male, $n = 6$ for female; male: treatment 1: $F_{(1,16)} = 8.230$, $p = 0.011$, $BF_{(incl)} = 3.005$; treatment 2: $F_{(1, 16)} = 4.926$, $p = 0.041$, $BF_{(incl)} = 1.380$; treatment 1 × treatment 2: $F_{(1,16)} = 4.680$, $p = 0.046$, $BF_{(incl)} = 1.737$; post-hoc comparisons: Veh + Saline vs Veh + CNO, $p = 0.013$; Veh + Saline vs 8-OH-DPAT+CNO, $p = 0.967$; female: treatment 1: $F_{(1, 20)} = 5.632$, $p = 0.028$, $BF_{(incl)} = 1.911$; treatment 2: $F_{(1, 20)} = 2.746$, $p = 0.113$, $BF_{(incl)} = 0.831$; treatment 1 × treatment 2: $F_{(1, 20)} = 4.320$, $p = 0.051$, $BF_{(incl)} = 1.714$; post-hoc comparisons: Veh + Saline vs Veh + CNO, $p = 0.024$; Veh + Saline vs 8-OH-DPAT+CNO, $p = 0.957$). (D) Effect of 8-OH-DPAT on social preference ratio in the three-chamber test ($n = 5$ for male, $n = 6$ for female; male: treatment 1: $F_{(1,16)} = 12.622$, $p = 0.003$, $BF_{(incl)} = 3.750$; treatment 2: $F_{(1,16)} = 4.993$, $p = 0.040$, $BF_{(incl)} = 0.987$; treatment 1 × treatment 2: $F_{(1,16)} = 12.622$, $p = 0.003$, $BF_{(incl)} = 10.906$; post-hoc comparisons: Veh + Saline vs Veh + CNO, $p < 0.001$; Veh + Saline vs 8-OH-DPAT+CNO, $p = 0.788$; female: treatment 1: $F_{(1, 20)} = 10.820$, $p = 0.004$, $BF_{(incl)}=6.321$; treatment 2: $F_{(1, 20)} = 3.075$, $p = 0.095$, $BF_{(incl)} = 0.858$; treatment 1 × treatment 2: $F_{(1, 20)} = 5.437$, $p = 0.030$, $BF_{(incl)} = 2.347$; post-hoc comparisons: Veh + Saline vs Veh + CNO, $p = 0.004$; Veh + Saline vs 8-OH-DPAT + CNO, $p = 0.702$). (E) Effect of 8-OH-DPAT on time spent in the central area in the open-field test ($n = 5$ for male, $n = 6$ for female; male: treatment 1: $F_{(1,16)} = 2.063$, $p = 0.170$, $BF_{(incl)} = 0.854$; treatment 2: $F_{(1,16)} = 0.759$, $p = 0.396$, $BF_{(incl)} = 0.523$; treatment 1 × treatment 2: $F_{(1,16)} = 0.003$, $p = 0.960$, $BF_{(incl)} = 0.498$; female: treatment 1: $F_{(1, 20)} = 0.144$, $p = 0.708$, $BF_{(incl)} = 0.393$; treatment 2: $F_{(1, 20)} = 0.324$, $p = 0.575$, $BF_{(incl)} = 0.422$; treatment 1 × treatment 2: $F_{(1, 20)} = 0.227$, $p = 0.639$, $BF_{(incl)} = 0.494$). (F) Effect of a 5-HT2AR antagonist (MDL 100907) on allogrooming time in the consolation test ($n = 6$ in each group; male: treatment 1: $F_{(1, 20)} = 37.118$, $p < 0.001$, $BF_{(incl)} = 1591.039$; treatment 2: $F_{(1, 20)} = 3.094$, $p = 0.094$, $BF_{(incl)} = 0.940$; treatment 1 × treatment 2: $F_{(1, 20)} = 2.697$, $p = 0.116$, $BF_{(incl)} = 1.101$; post-hoc comparisons: Veh + Saline vs Veh + CNO, $p < 0.001$; Veh + Saline vs MDL 100907 + CNO, $p<0.001$; female: treatment 1: $F_{(1, 20)} = 36.168$, $p<0.001$, $BF_{(incl)} = 2649.973$; treatment 2: $F_{(1, 20)} = 0.860$, $p = 0.365$, $BF_{(incl)} = 0.484$; treatment 1 × treatment 2: $F_{(1, 20)} = 1.554$, $p = 0.227$, $BF_{(incl)} = 0.780$; post-hoc comparisons: Veh + Saline vs Veh + CNO, $p<0.001$; Veh + Saline vs MDL 100907 + CNO, $p < 0.001$). (G)

*Figure 6 continued on next page*

*Figure 6 continued*

Effect of MDL 100907 on social preference ratio in the three-chamber test ($n = 6$ in each group; male: treatment 1: $F_{(1, 20)} = 26.890$, p<0.001, $BF_{(incl)} = 771.577$; treatment 2: $F_{(1, 20)} = 0.124$, p = 0.729, $BF_{(incl)} = 0.386$; treatment 1 × treatment 2: $F_{(1, 20)} = 0.316$, p = 0.580, $BF_{(incl)} = 0.496$; post-hoc comparisons: Veh + Saline vs Veh + CNO, p = 0.003; Veh + Saline vs MDL 100907 + CNO, p = 0.004; female: treatment 1: $F_{(1, 20)} = 13.991$, p = 0.001, $BF_{(incl)} = 34.866$; treatment 2: $F_{(1, 20)} = 0.620$, p = 0.440, $BF_{(incl)} = 0.443$; treatment 1 × treatment 2: $F_{(1, 20)} = 0.071$, p = 0.793, $BF_{(incl)} = 0.500$; post-hoc comparisons: Veh + Saline vs Veh + CNO, p = 0.047; Veh + Saline vs MDL 100907 + CNO, p = 0.191). (H) Effect of MDL 100907 on time spent in the central area in the open-field test ($n = 6$ in each group; male: treatment 1: $F_{(1, 20)} = 3.702$, p = 0.069, $BF_{(incl)} = 1.469$; treatment 2: $F_{(1, 20)} = 0.415$, p = 0.527, $BF_{(incl)} = 0.429$; treatment 1 × treatment 2: $F_{(1, 20)} = 0.270$, p = 0.609, $BF_{(incl)} = 0.520$; female: treatment 1: $F_{(1, 20)} = 1.506$, p = 0.234, $BF_{(incl)} = 0.655$; treatment 2: $F_{(1, 20)} = 0.274$, p = 0.606, $BF_{(incl)} = 0.417$; treatment 1 × treatment 2: $F_{(1, 20)} = 1.305$, p = 0.267, $BF_{(incl)} = 0.698$). Two-way ANOVA along with two-way Bayesian ANOVA. Error bars are ± SEM. *p < 0.05, **p < 0.01. For raw data in this figure, please refer to *Figure 6—source data 1*. Anta: antangonist; ACC: anterior cingulate cortex; ANOVA: analysis of variance; DR: dorsal raphe nucleus; 5-HT: serotonin.

The online version of this article includes the following source data and figure supplement(s) for figure 6:

**Source data 1.** Source data indicating 8-OH-DPAT but not MDL 100907 reduced sociability deficits induced by chemogenetic inhibition of DR 5-HT neurons in the DR→ACC neural circuit.

**Figure supplement 1.** Comparisons of 'Vehicle + Saline - Vehicle + CNO' vs '8-OH-DPAT + Saline - 8-OH-DPAT + CNO' (A and B) and 'Vehicle + Saline - Vehicle + CNO' vs 'MDL 100907 + Saline MDL 100907 + CNO' (C and D).

**Figure supplement 1—source data 1.** Source data indicating pretreatment with 8-OH-DPAT but not MDL 100907 significantly reduced the impact of CNO.

**Figure supplement 2.** Chemogenetic inhibition of DR 5-HT neuron in the DR-ACC neural circuit along with intra-ACC.

**Figure supplement 2—source data 1.** Source data indicating the treatments had no significant effect on control behaviors of selfgrooming and the distance traveled in the open-field test.

## Discussion

In the present study, we have demonstrated a crucial role for the DR→ACC 5-HTergic neural circuit in the regulation of consolation-like behaviors and sociability in mandarin voles for the first time. Our major findings are listed below. First, inhibition of DR 5-HT neurons or their terminals in the ACC decreased allogrooming behavior and reduced sociability. Second, DR 5-HT neuron activity and ACC 5-HT release increased during allogrooming, social approaching, and sniffing. Third, direct activation of ACC 5-HT1A receptors was sufficient to ameliorate deficits in consolation and sociability induced by chemogenetic inhibition of DR 5-HT neurons.

We found that optogenetic inhibition of DR 5-HT neurons (*Figure 1*) or ACC 5-HT terminals (*Figure 2*) significantly decreased intimate behaviors (allogrooming and chasing) toward their distressed partners and reduced sociability. Chemogenetic inhibition of DR 5-HT neurons produced similar results (*Figure 3*). Consistent with our results, *Walsh et al., 2018* found that optogenetic inhibition of DR 5-HT neurons reduced social interactions in the three-chamber test. As consolation is in general a pro-social behavior, the reduced allogrooming may be largely due to an overall decrease in sociability. Further experiments are needed to address this interesting question.

Another open question is how this process occurs, namely, what the neural mechanisms underlying this process are. The neocortical excitatory/inhibitory (E/I) balance hypothesis may help address this question. The hypothesis indicates that an increase in the cortical cellular E/I balance, for example, through increased activity in excitatory neurons or a reduction in inhibitory neuron function, is the common etiology and final pathway for some psychiatric diseases, such as autism and schizophrenia (*Bozzi et al., 2018*; *Vattikuti and Chow, 2010*). This hypothesis has recently been verified in mice, as optogenetic excitation of glutamatergic neurons in the mPFC elicits a profound impairment in sociability, while compensatory excitation of inhibitory neurons in this region partially rescues the social deficits caused by an increase in the E/I balance (*Yizhar et al., 2011*). 5-HT tends to inhibit prefrontal pyramidal activity (*Puig and Gulledge, 2011*; *Tian et al., 2017*). For example, according to *Puig et al., 2005*, electrical stimulation of the DR inhibits approximately two-thirds of pyramidal neurons in the mPFC. Therefore, a reasonable hypothesis is that inhibition of the DR-ACC 5-HTergic neural circuit may increase the E/I balance, which ultimately leads to abnormal social

behaviors in mandarin voles. Clearly, this hypothesis should be verified in electrophysiological studies in the future.

In contrast to inhibition manipulations, we found that activation of the DR-ACC 5-HTergic neural circuit did not elicit corresponding increases in allogrooming and sociability (*Figures 1–3*). One possible explanation is that there are various 5-HT receptors expressed in the ACC and some have opposite functional effects (*Puig and Gulledge, 2011*; *Tian et al., 2017*). For example, 5-HT1AR coupled to the Gi family of G proteins induces hyperpolarization of pyramidal neurons, whereas 5-HT2AR coupled to Gq proteins induces depolarization in the same neurons (*Carhart-Harris and Nutt, 2017*). In addition, these two types of receptors are expressed in both pyramidal and GABAergic neurons of the mPFC (*Santana et al., 2004*). Therefore, the net behavioral effects of 5-HT release must result from the combined effects of all these receptors. The other possible explanation is a ceiling effect, which means that the enhancement of consolation and sociability is difficult to accomplish in these normal animals. This may occur in situations where levels of consolation and sociability are low, such as in the stressed subjects in our previous study (*Li et al., 2020*) or in the genetically less-sociable subjects as in the study by *Walsh et al., 2018*. However, in a previous study by *Kim et al., 2014*, the direct administration of 5-HT into the ACC impaired observational fear learning, which is an indicator of emotional contagion. Species differences (mice vs voles), methodological differences (pharmacology vs optogenetics and chemogenetics) or different behavioral indicators (consolation vs emotional contagion) may account for the discrepancies. Furthermore, optogenetic or chemogenetic activation of DR 5-HT neurons is also distinct from pharmacological intervention with MDMA, which robustly induces 5-HT releases in the whole brain and enhances closeness and empathy in its users in human studies (*Carlyle et al., 2019*; *Heifets and Malenka, 2016*). Nevertheless, at least some effects of activation were visible. For example, optogenetic activation of ACC 5-HT terminals increased sociability in CHR2-expressing females (*Figure 2E*). Clearly, the effects of activation of the DR-ACC 5-HTergic neural circuit on sociability and empathy still require further in-depth study.

Our results show that neither activation nor inhibition of the DR-ACC 5-HTergic neural circuit exerts any significant effect on some control behaviors (*Figure 1—figure supplement 4*; *Figure 2—figure supplement 1*; *Figure 3—figure supplement 2*), that is, selfgrooming in the consolation test or behavioral performance in the open-field test, which is consistent with previous results (for selfgrooming, please refer to *Correia et al., 2017*; for the open-field test, please refer to *Walsh et al., 2018*). However, inconsistent results have also been reported. For example, *Ohmura et al., 2014* found that optogenetic activation of 5-HT neurons in the median raphe nucleus enhanced anxiety-like behaviors in mice. In the study by *Correia et al., 2017*, activation of 5-HTergic neurons in the DR affected behavior in the open-field test, but not anxiety. Different stimulation protocols or manipulations of different groups of 5-HT neurons may account for these discrepancies.

Our results obtained using in vivo fiber photometry indicated an increase in DR 5-HT neuron activity during allogrooming, sniffing, and social approaching (*Figure 4*). This result is consistent with previous studies showing increases in the activity of 5-HTergic neurons in the DR during nonaggressive social interactions (*Li et al., 2016*; *Walsh et al., 2018*). Furthermore, using the highly sensitive 5-HT fluorescent sensors developed by *Wan et al., 2021*, we provided the first evidence that allogrooming, sniffing, and social approaching elicit robust 5-HT release in the ACC (*Figure 5*). Selfgrooming and running did not induce robust 5-HT release within the ACC, which may indicate the behavioral relevance of the 5-HT sensors. We also found that the fluorescence changes in GCaMP6 and 5-HT sensors usually dropped before the end of these social behaviors (*Figure 4—figure supplement 2*, *Figure 5—figure supplement 1*). These results indicate that 5-HT may be involved in some motivational aspects of these social behaviors, which is consistent with previous findings (*Arakawa, 2020*; *Yagishita, 2020*).

Although both 5-HT1AR and 5-HT2AR are expressed at high levels in the mPFC and work together to modulate cortical network activity (*Celada et al., 2013*), we found only the 5-HT1AR receptor agonist significantly reduced the consolation and sociability deficits induced by chemogenetic inhibition of 5-HTergic neurons in the DR (*Figure 6*). This result is consistent with our previous finding that the administration of WAY-100635 (a 5-HT1AR antagonist) into the ACC attenuates consolation and sociability in mandarin voles (*Li et al., 2020*). 5-HT1AR within the mPFC has frequently been reported to exert antidepressant and anxiolytic effects (*Artigas, 2013*; *Fukumoto et al., 2020*; *Fukumoto et al., 2014*; *Wang et al., 2019*), but our findings clearly indicate that this type of 5-HT

receptor within the ACC is also involved in regulating some social behaviors. Clearly, this effect should be verified in many other species in future studies. Administration of the 5-HT2AR antagonist into the ACC did not exert an obvious effect on the social deficits induced by the chemogenetic inhibition of DR 5-HT neurons (*Figure 5F–H*). Thus, 5-HT2AR within the ACC may not play a major role in consolation and sociability. However, this conclusion should be interpreted very cautiously because 5-HT2AR expression within the mPFC shows clear rostral-to-caudal gradients in mice (*Weber and Andrade, 2010*) and we were unable to determine its functional role in this dimension in the present study. Furthermore, the effects of 5-HT in the ACC on consolation and sociability probably involve other subtypes of 5-HT receptors, such as 5-HT1BR and 5-HT3R, which requires further examination.

In conclusion, our findings establish the importance of the DR-ACC 5-HTergic neural circuit in consolation-like behaviors and sociability in mandarin voles. One major limitation of our study is that although our virus strategies were generally successful, a few of the infected neurons were indeed TPH2-negative, which may challenge the specificity of our manipulations and confound our results. However, the fiber photometry and pharmacological studies mitigate these concerns. Considering the widespread innervation of 5-HT terminals and abundant distribution of 5-HT receptors in the whole brain, detailed knowledge of cellular and circuit mechanisms of the 5-HTergic system will not only improve our understanding of its complicated features and functions but also have implications for the development of novel therapies for the treatment of prevalent neuropsychiatric disorders, such as depression, autism, and schizophrenia. Additionally, although both male and female subjects were included in this study, sexually dimorphic effects were rarely observed. This finding may provide additional evidence of cooperative evolution to adapt to environmental challenges, particularly in species that adapt to monogamous relationships and require cooperative breeding.

# Materials and methods

**Key resources table**

| Reagent type (species) or resource | Designation | Source or reference | Identifiers | Additional information |
|---|---|---|---|---|
| Antibody | Anti-TPH2 (goat polyclonal antibody) | Abcam | ab121013; RRID:AB_10898794 | IF (1/500) |
| Antibody | Alexa Fluor 488-AffiniPure Donkey Anti-Goat IgG (H + L) | Jackson | 705-545-147 RRID:AB_2336933 | IF (1/300) |
| Antibody | Rhodamine (TRITC)-AffiniPure Donkey Anti-Goat IgG (H + L) | Jackson | 705-025-003 RRID:AB_2340388 | IF (1/300) |
| Chemical compound, drug | 8-OH-DPAT | Sigma | H8520 | Injection site: ACC; volume: 200 nl/side; concentration:1.5 mg/ml |
| Chemical compound, drug | MDL 100907 | Tocris Bioscience | Cat# 4173 | Injection site: ACC; volume:200 nl/side; concentration:1 mg/ml |
| Chemical compound, drug | Metergoline | MedChemExpress | Cat# HY-B1033/ CS-4551 | 4 mg/kg, i.p. |
| Chemical compound, drug | CNO | BrainVTA | Cat# CNO-02 | 1 mg/kg, i.p. |
| Chemical compound, drug | Cholera Toxin Subunit B (CTB)594 | Thermo Fisher | Cat# C34777 | Injection site: ACC; volume:400 nl/side; expression: 10 days |
| Transfected construct (human) | rAAV(Retro)-TPH2-Cre-WPRE-pA | BrainVTA | R-396-K181127 | Serotype: AAV2; titer: 5.86 × 10$^{12}$ vg/ml; injection site: ACC; volume: 300 nl/side |

*Continued on next page*

*Continued*

| Reagent type (species) or resource | Designation | Source or reference | Identifiers | Additional information |
|---|---|---|---|---|
| Transfected construct (human) | rAAV-Ef1α-DIO-ChR2-mCherry-WPRE-pA | BrainVTA | 9–2 K190827 | Serotype: AAV2/9; titer: $2.01 \times 10^{12}$ vg/ml; injection site: DR; volume: 500 nl |
| Transfected construct (human) | rAAV-Ef1α-DIO-eNpHR3.0-mCherry-WPRE-pA | BrainVTA | rAAV9-7-5-1 | Serotype: AAV2/9; titer: $2.23 \times 10^{12}$ vg/ml; injection site: DR; volume: 500 nl |
| Transfected construct (human) | rAAV-Ef1α-DIO-mCherry-WPRE-pA | BrainVTA | 9–13 K190430 | Serotype: AAV2/9; titer: $2.02 \times 10^{12}$ vg/ml; injection site: DR; volume: 500 nl |
| Transfected construct (human) | rAAV-Ef1α-DIO-hM4Di(Gi)-mCherry-WPRE-pA | BrainVTA | 9–43 K190521 | Serotype: AAV2/9; titer: $2.09 \times 10^{12}$ vg/ml; injection site: DR; volume: 500 nl |
| Transfected construct (human) | rAAV-Ef1α-DIO-hM4Dq(Gq)-mCherry-WPRE-pA | BrainVTA | 9–42 K190530 | Serotype: AAV2/9; titer: $2.70 \times 10^{12}$ vg/ml; injection site: DR; volume: 500 nl |
| Transfected construct (human) | rAAV-hSyn-5HT2.1-WPRE-hGHpA (5-HT sensor) | BrainVTA | 9–1826 K190620 | Serotype: AAV2/9; titer: $2.53 \times 10^{12}$ vg/ml; injection site: ACC; volume: 400 nl |
| Transfected construct (human) | rAAV-hSyn-EGFP-WPRE-hGH-pA | BrainVTA | PT-1990 | Serotype: AAV2/9; titer: $2.00 \times 10^{12}$ vg/ml; injection site: ACC; volume: 400 nl |
| Transfected construct (human) | rAAV-EF1α-DIO-GCaMp6-WPRE-hGH pA | BrainVTA | PT-0071 | Serotype: AAV2/9; titer: $5.15 \times 10^{12}$ vg/ml; injection site: DR; volume: 500 nl |
| Transfected construct (human) | rAAV-hSyn-DIO-EGFP-WPRE-hGH-PA | BrainVTA | 9–1103 K200409 | Serotype: AAV2/9; titer: $2.00 \times 10^{12}$ vg/ml; injection site: DR; volume: 500 nl |
| Software, algorithm | JASP | JASP | RRID:SCR_015823 | |
| Other | DAPI stain | Boster | AR1177 | 1 µg/ml |

## Animals

The mandarin voles used in this study were laboratory-bred strains (F2-F3) whose ancestors were derived from a wild population from Lingbao city (Henan, China). The voles were weaned on postnatal day 21, socially housed in same sex in separate polycarbonate cages (44 × 22 × 16 cm), and housed on a 12 hr light/dark cycle with food and water ad libitum. Voles used for experiments were about 70–90 days old at the time. All breeding, housing, and experimental procedures were in accordance with Chinese guidelines for the care and use of laboratory animals and were approved by the Animal Care and Use Committee of Shaanxi Normal University.

## Stereotaxic surgery and virus infusions

The kind of virus, total injection volumes, and the expression time are listed in Key Resources Table. For surgery, voles (about 50 days) were anesthetized with 1.5–3.0% isoflurane and placed in a stereotaxic instrument. Next, 33-gauge syringe needles (Hamilton) were used for virus delivery. The injection rates were set at 50 nl/min. After each injection, the needle was left in the brain for another 5 min before being slowly withdrawn in order to prevent the virus from leaking out. The bregma

coordinates for the virus injection were as follows: ACC: A/P: 1.6, M/L: 0.5, D/V: 1.6; DR: AP: −4.3; DV: −3.3, ML:+1.2, with a 20° angle toward the midline in the coronal plane.

## Microinjection and drugs

The 5-HT1AR agonist 8-OH-DPAT was prepared in saline with a final concentration of 1.5 mg/ml. The 5-HT2AR antagonist MDL 100907 was prepared in 0.01 M phosphate-buffered saline (pH value adjusted to 6.4 with 0.1 M HCl) with a final concentration of 1 mg/ml. All the microinjections were administered 30 min before the behavioral test. The speed of injection was 0.1 μl/min, and the total injection volume was 0.2 μl per side for all the drugs. The injector tips remained in situ for another 2 min for drug diffusions. The dose and timing of drug administration were chosen based on previous studies with 8-OH-DPAT (*Cooper et al., 2008*; *Li et al., 2020*) and MDL 100907 (*Ishii et al., 2015*; *Pockros et al., 2011*), which were adjusted according to preliminary studies. In chemogenetic studies, CNO (1 mg/kg) was dissolved in saline and delivered i.p. 30 min before the behavioral test.

## Optogenetic studies

For optogenetic activation, ferrules were connected (by patch cords) to a 473 nm laser diode through an FC/PC adaptor and a fiber optic rotary joint. The output parameters were 10 ms, 20 Hz, 8 s on and 2 s off cycle, ~10 mW for terminal stimulation, ~5 mW for somatic stimulation. For optogenetic inhibition, ferrules were connected to a 593 nm laser diode. The output parameters were 10 ms, 20 Hz, constant, ~10 mW for both terminal and somatic inhibition. The optogenetic stimulation parameters were chosen and slightly modified based on previous studies (*Garcia-Garcia et al., 2018*; *Walsh et al., 2018*; *Zhao et al., 2011*).

## Behavioral assays

Generally, all the behavioral experiments were performed under dim light during the dark phase of the light-dark cycle. In all tests, all groups of experimental voles were randomly selected and the observers were blinded to the treatments. During the test, if the subjects (not the stimulus voles) showed very rare or no movements, these individuals were not expected to show normal activity, and their data were not appropriate to include in the following analysis (*Figure 1I, L* and *Figure 2H*). For optogenetic and fiber photometry studies, all subjects were allowed to habituate to patch cords for at least 3 days and allowed 30 min acclimation to the connection before the experiment started.

## Consolation test

The consolation test was performed as previously described (*Burkett et al., 2016*). Briefly, 5 days before the experiment, age-matched adult male and female voles were cohoused together to promote the formation of a pair bond (*Yu et al., 2012*). On the testing day, the subjects' partners were gently transferred in a cup to a sound-proof electric shock chamber and then subjected to 10 rounds of moderately strong foot shocks (3 s, 0.8 mA, 2-min intertrial intervals). At the same time, the test voles were left undisturbed in their home cages. After the separation, the pairs were reunited, and the behavior of the subjects was recorded using a video camera for 10 min in the test room. The digital videos were viewed and quantified using J Watcher software (http://www.jwatcher.ucla.edu/). According to previous studies (*Burkett et al., 2016*; *Li et al., 2019*), allogrooming is designated as behavioral indicators of consolation, which is defined as head contact with the body or head of their partner, accompanied by a rhythmic head movement. The duration of chasing (following the shocked partners) and selfgrooming was also collected and analyzed.

Optogenetic test was designed as a between-subject test in which both groups (mCherry or opsin) received laser stimulation. The lasting effects of optogenetic inhibition were measured 24 hr after the original test, that is, consolation test was conducted once again without light stimulation in eNPHR 3.0 animals 1 day later.

## Three-chamber test

The three-chamber test was used to assess the sociability of a subject (*Horie et al., 2019*; *Walsh et al., 2018*). The apparatus consisted of a rectangular box with three separate chambers (20 × 40 × 20 cm each). One side of each chamber contained a circular metal wire cage (stimulus

animal cage, 11 cm high and 9 cm in diameter). 1 day before the test, all subjects were habituated to the arena for 5 min and age- and sex-matched unfamiliar stimulus voles were also habituated to the wire cages for 5 min. On the testing day, the test vole was placed in the center chamber and a 'stimulus' vole (housed in the same room with the test voles) was randomly placed into one of the wire cages. After 2 min, the partitioning walls between the chambers were removed and the test voles were allowed to explore freely for a 10 min session. The time spent exploring each chamber was automatically recorded using a video tracking system (Shanghai Xinruan, China). Sociability was calculated as follows: (time spent in the stimulus vole side − time spent in the empty side)/(time spent in the stimulus vole side + time spent in the empty side).

For optogenetic tests, voles received four epochs of light beginning with a light OFF baseline epoch (OFF-ON-OFF-ON). Each epoch lasted for 5 min and the total duration was 20 min.

## Open-field test

The anxiety level and locomotor function of the subjects were assessed by using the open-field test (*Flanigan et al., 2020*; *Walsh et al., 2018*). Briefly, a square open field (50 cm × 50 cm × 25 cm) was virtually subdivided into 16 even squares. The four central squares were designated as central area. At the beginning of the test, the test vole was placed into the center area facing away from the experimenter. Behavior was recorded for 5 min. Outcome measures were distance traveled, frequency entries, and time spent in the central area.

For optogenetic tests, voles were tested twice on the same day in both light-off and light-on conditions with at least 2 hr between sessions (within-subject design).

## Fiber photometry

The fiber photometry recording set-up (ThinkerTech, Nanjing, China) was generated and used as previously described (*Feng et al., 2019*; *Yuan et al., 2019*). Briefly, the emission light was generated by a 480 LED, reflected with a dichroic mirror, and delivered to the brain in order to excite the GCaMP6m or the 5-HT sensor. The emission light was passed through another band pass filter, into a CMOS detector (Thorlabs, Inc; DCC3240M), and finally recorded by a Labview program (TDMSViewer; ThinkerTech, Nanjing, China).

On the test day, voles were mildly anesthetized with isoflurane and connected to a multi-mode optic fiber patch cord (ThinkerTech Nanjing Bioscience; NA: 0.37, OD: 200 µm) whose other end was connected to a fiber photometry apparatus. After 30-min habituation, the voles were subjected to consolation test as described above (exposure to their shocked partners), and the behavioral recording consisted of allogrooming, sniffing, social approaching, selfgrooming, and running. Four to six bouts of a behavior were collected during the test. If a specific behavior occurred no more than four times within 1 hr, the record was replicated on the following day.

Fiber photometry signals were processed with a custom-written MATLAB software, which is available at https://github.com/ganenlife1980/MultiFiberPhotometry__MetLab (copy archived at swh:1:rev:e89f3a67bb9d40d5e82017f895b2bed74f17a42c; *ThinkerTech Nanjing Bioscience, 2020*). Briefly, all the data were segmented based on the behavioral events and baseline phase. The change in fluorescence (ΔF/F) was calculated as (F–F0)/F0, where F0 represents the baseline fluorescence signal averaged over a 10 s-long control time window. We first segmented the data based on the behavioral events. Then, we calculated the average 5-HT and calcium signals in both the pre- and post-phases (2/8 s). The time window was determined according to the duration of a behavior bout. The response elicited during a behavior was calculated as the average ΔF/F during all trials of a particular behavior.

For specificity verification of 5-HT sensors in *Figure 5—figure supplement 2*, a 5-HT receptor antagonist metergoline (Met) was injected (i.p., 4 mg/kg) 20 min before the fiber photometry experiments. The timing and dosing were chosen according to previous studies (*Onasanwo et al., 2016*; *Wan et al., 2021*).

## In vitro electrophysiological recordings

To verify the optogenetic and chemogenetic manipulations, we performed in vitro whole-cell patch-clamp recordings from DR neurons. Neurons expressing ChR2, eNpHR3.0, hM3Dq, and hM4Di were visually identified by fluorescence of mCherry. The voles were anesthetized with isoflurane. Brains

were quickly dissected and 300 µm coronal slices containing the DR were prepared in a chamber filled with artificial cerebrospinal fluid (ACSF) (32–34°C) using vibratome (Campden 7000 smz). The recordings were obtained using a Multiclamp 700B amplifier, filtered at 5 kHz, and sampled at 10 kHz with Digidata 1440A. Clampex 10.5 was used for analysis.

Current-clamp recordings were performed to measure evoked action potentials. For photoactivation and photoinhibition, the light protocols used during behavioral tests were delivered through a 200 µm optical fiber close to recorded neurons. For CNO activation, spontaneous firing of action potentials in the cell was recorded in the current-clamp mode at −60 mV. After 5 min of recording, the slices were perfused with 10 µM CNO. The total recording time for each cell was 10 min. For CNO inhibition, we applied currents in steps of 25 pA, ranging from 0 pA to 250 pA. Neurons were allowed to recover for 10 min and then perfused with 10 µM CNO. The same current procedure was performed. Afterwards, the CNO was removed by washes with ASCF and cells were recorded for another 10 min.

## Data analysis

All data are represented as the mean ± SE. All data were assessed for normality using a one-sample Kolmogorov-Smirnov test, and the Levene's test was used to confirm homogeneity of variance. Comparisons between two groups were performed by unpaired or paired $t$-tests. Comparisons among three or more groups of different animals were performed using one-way ANOVA. Two-way ANOVAs or two-way repeated measures ANOVAs were used to compare multiple groups under multiple testing conditions when appropriate. Post-hoc comparisons were conducted using Tukey. To confirm the significance and calculate the effect size, Bayesian $t$-test (paired or unpaired) or Bayesian ANOVA were conducted for all the comparisons using default priors. For more detailed information about the Bayesian test, please refer to *Keysers et al., 2020*. The data were analyzed using JASP 14.0 and are presented as the mean ± SE. The significant level was set at p < 0.05. The detailed analysis method in each figure and the statistical quantifications are presented in *Figure 2— figure supplement 2*.

## Acknowledgements

We thank Yu-Ying Yang, Xin Zhang, and Yi-Xin Feng for assistance in conducting experiments and caring for voles. The authors declare no conflict of interest.

## Additional information

### Funding

| Funder | Grant reference number | Author |
| --- | --- | --- |
| National Natural Science Foundation of China | 31970424 | Fa-Dao Tai |
| Natural Science Foundation of Shaanxi Province | 2018JM3032 | Fa-Dao Tai |
| Fundamental Research Funds for the Central Universities | GK201903059 | Fa-Dao Tai |
| National Natural Science Foundation of China | 31670421 | Fa-Dao Tai |
| National Natural Science Foundation of China | 31372213 | Fa-Dao Tai |
| National Natural Science Foundation of China | 31772473 | Fa-Dao Tai |

The funders had no role in study design, data collection and interpretation, or the decision to submit the work for publication.

## Author contributions
Laifu Li, Investigation, Writing - original draft, Project administration; Li-Zi Zhang, Data curation, Methodology; Zhi-Xiong He, Supervision, Investigation, Methodology; Huan Ma, Yu-Ting Zhang, Investigation, Visualization, Methodology; Yu-Feng Xun, Validation, Investigation, Visualization; Wei Yuan, Yi-Tong Li, Validation, Investigation; Wen-Juan Hou, Investigation, Visualization; Zi-Jian Lv, Rui Jia, Supervision, Visualization; Fa-Dao Tai, Resources, Supervision, Writing - review and editing

## Author ORCIDs
Fa-Dao Tai  https://orcid.org/0000-0002-6804-4179

## Ethics
Animal experimentation: All the breeding, housing, and experimental procedures in this study were in accordance with Chinese guidelines for the care and use of laboratory animals and were approved by the Animal Care and Use Committee of Shaanxi Normal University (SNNU_20190501001). All efforts were made to minimize suffering and the number of animals used during the studies.

## Decision letter and Author response
Decision letter https://doi.org/10.7554/eLife.67638.sa1
Author response https://doi.org/10.7554/eLife.67638.sa2

# Additional files

## Supplementary files
• Transparent reporting form

## Data availability
All data generated or analysed during this study are included in the manuscript and supporting files. We have also deposited the datasets of this manuscript into Dryad.

The following dataset was generated:

| Author(s) | Year | Dataset title | Dataset URL | Database and Identifier |
|---|---|---|---|---|
| Tai F, Li L, Zhang L-Z, He Z-X, Zhang Y-T, Ma H, Xun Y-F, Yuan W, Hou W-J, Li Y-T, Lv Z-J, Jia R | 2021 | Dorsal raphe nucleus to anterior cingulate cortex 5-HTergic neural circuit modulates consolation and sociability | http://dx.doi.org/10.5061/dryad.8931zcrq7 | Dryad Digital Repository, 10.5061/dryad.8931zcrq7 |

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
