## [Decision Letter]

**Acceptance summary:**

While the anterior cingulum is implicated in many affective and social behaviors, this study breaks ground in implicating 5ht dorsal raphe neurons in the same. Given the rich pharmacopeia of drugs acting on 5ht transmission, the findings hold translational promise.

**Decision letter after peer review:**

[Editors’ note: the authors submitted for reconsideration following the decision after peer review. What follows is the decision letter after the first round of review.]

Thank you for submitting your work entitled "Dorsal raphe nuclei to anterior cingulate cortex 5-HTergic neural circuit modulate consolation and sociability" for consideration by *eLife*. Your article has been reviewed by 4 peer reviewers, , including Peggy Mason as the Reviewing Editor and Reviewer #1, and the evaluation has been overseen a Senior Editor. The following individuals involved in review of your submission have agreed to reveal their identity: Christian Keysers (Reviewer #2); JP Burkett (Reviewer #3).

This is a multi-pronged study of the contribution of 5ht projections from DR to ACC in consolation behavior in mandarin voles. There was much that the reviewers liked about this study and the manuscript. However, concerns about the photometry data lead us to reject the manuscript, in its present form. There are concerns about the specificity of the method for detecting 5ht. These concerns, well articulated by Reviewer 4 and described below, require new experiments. Should you be able to gather data that assuage these concerns, we would be willing to consider this manuscript as a new submission. While specific reviews are included below, I also highlight here several other concerns that will be critical to address for a successful revision.

– Photometry data: We noted that recordings of the 5HT sensor are confounded by the potential presence of GCAMP in ACC terminals. This could be resolved in several ways for example conducting the 5HT and GCAMP recordings in separate subjects. Validation for the selectivity of the 5HT sensor is needed. A control for motion artifact is also needed.

– Is the observed effect attributable specifically to consolation or could it be a more general effect on sociability? The latter possibility should be considered and discussed.

– The exclusion of data from animals that froze is highly problematic. Please expand on how often this exclusionary criterion was invoked and provide justification for the exclusion.

– The statistical tests used were not compelling to the reviewers. Reviewer #2 outlines many ways to improve this aspect of the presentation. In addition, the authors should consider straightforward equivalence testing, such as equivalence tests (eg TOST) as well as ANOVAs (and MANOVAs) to address correcting for multiple comparisons (see Reviewer 3).

– The change in fluorescence follows the behavior of interest. This is not compatible with the causality implied by the authors' interpretation. This needs to be discussed and possibly aspects of the timing re-analyzed along the lines suggested by Reviewer #2.

– Is consolation behavior at a maximum such that a further enhancement could not be observed with 5ht? The possibility of a ceiling effect as outlined by Reviewer #2 should be considered.

– A schematic showing histological verification of all fiber placements for each experiment is needed. In addition to showing microscopy images showing a larger region of the brain to confirm the image location, it would also help to show a schematic representation of the viral expression spread for each subject (e.g., overlaid on brain slice schematics) for each experiment.

*Reviewer #1:*

This is a multi-pronged study of the contribution of 5ht projections from DR to ACC in consolation behavior in mandarin voles.

There are writing issues that make less than facile to read. However, these issues can be overcome, and the data are extensive and worthy of consideration despite the challenges to the reader.

The one critical concern that I have is that the authors have not demonstrated that this is consolation behavior as opposed to reunification behavior. In the Burkett et al. study, a very important control of separation without shock was performed. Here that control was not performed. Thus fundamentally the argument that what is being represented here is reunion behavior rather than consolation cannot be refuted by the available evidence.

The behavior should be described in the Results as well as in the methods. *eLife* is a journal for the general reader who is unlikely to be familiar with the empathy literature.

Are there 5ht terminals in the ACC from median raphe?

Make the scales in Figure 4 constant.

293 Why put a 5ht2ar antagonist while inhibiting DR 5ht neurons? Why not put in an agonist as was done with the 5ht1ar?

*Reviewer #2:*

In their paper, Li et al., examine the contribution of serotonin in the ACC to consolation behavior in voles. They show that consolation (as measured as an increase of allogroming towards a mate that has been exposed to shocks), as well as social preference (as measured as more time spent close to a conspecific in a 3 compartment test), are reduced when inhibiting the dorsal raphe->anterior cingulate 5HT pathway, and that this can be rescued using a 5HT1A receptor agonist. In addition, they measure transient increases of 5HT in the ACC following the onset of allogrooming and social approach.

In general, this is an interesting and well-conducted study, that combines techniques to interfere with the DR->ACC pathway (including chemo and optogenetic approaches) with some physiology (calcium imaging in DR and 5HT sensors in ACC), to create a series of studies that compel the reader to accept the importance of 5HT on allogroming and social approach.

a) Statistic reporting. The paper contains a large number of data, which is great. However, the presentation of the statistical tests and results is often difficult to understand in detail. I applaud the presence of the data-tables for download, and the rigorous illustration of single subject data in most figures. However, results in the text seldom allow to fully understand what statistical test was used. For instance, in the 3 chamber test in Figure 1, the appropriate test would appear to be a group (mcherry vs Chr2) x condition (laser on vs off) interaction in an ANOVA, which is equivalent to a two-sample t-test of the difference On – Off. Please specify in the text and the figures whether an ANOVA looked at this interaction specifically, and always report exact p-values. Also in the figures, when specific t-test were performed, please indicate the test statistic in the figure (e.g. t(n)=x, p=0.0x), whether significant or not, given there is enough space in the figures. In the current version it is often difficult to assess whether statements of effect or no effect are based on ANOVAs (and if so on what interaction or main effect in that ANOVA) or on posthoc t-tests. I know this makes figures a bit more cluttered, but it provides the reader with a richer understanding of the data.

b) Bayesian statistics. The manuscript contains many highly relevant statements that hinge on the fact that a particular condition had no effect on a particular outcome measure, or that a condition brings contagion levels back to normal. These are statements that frequentist statistics (t-test or ANOVA) cannot support, but that are important in the paper. This is because p>0.05 does not provide evidence for H0 (the absence of an effect). Bayes Factor analyses, in contrast, can easily test (see Keysers, Gazzola and Wagenmakers, Nature Neuroscience, 2020 for details of why and how to do that) and quantify evidence for H0. Therefore please always also provide the Bayes factor for the t-tests, and the Bayesfactor for the inclusion of a factor/interaction in an ANOVA. Given that most of your hypotheses are directional, e.g. inhibition of DR->ACC should reduce allogrooming and social preference, I would recommend using directed hypotheses for the Bayesian test to provide more power in favour of H0. This will indicate whether you have evidence that a manipulation had no effect (BF-0<1/3 or BFincl<1/3) or whether you simply have too much variance or two small a group size to establish that a condition had no effect (1/3<BF<3). These analyses can be easily done using JASP on the tables that you provide, and please indicate the BF-0 values together with the t-values in your figures, and adjust your result description according to whether you have evidence of absence or absence of evidence. I know this is some additional work, and people are only starting to adopt BF testing, but this would really help the reader understand when you really have evidence that a manipulation had no effect (BF<1/3) vs. cases in which your group size is simply too small to exclude an effect (see power considerations in point c).

c) t-test vs ANOVA: in some cases, your statements call for ANOVA's or Bayesian statistics not t-tests. For instance you write: "Pretreatment with 8-OH-DPAT reversed the dysfunctions to the normal level (all P > 0.05 for '8-OH-DPAT+CNO' vs. 'vehicle+saline' pos-hoc comparisons; Figure 5C-D)". Now, to state that it fully reversed to normal level, you need to show that BF-0 for '8-OH-DPAT+CNO' vs. 'vehicle+saline' is <1/3. If you lack power to establish that, you could use an ANOVA to show that 'Vehicle+Saline – Vehicle+CNO > 8-OH-DPAT+Saline – 8-OH-DPAT+CNO', and say that 8-OH-DPAT+CNO significantly reduces the impact of CNO. Both of these would be evidence for something. Currently, simply showing the lack of a significant effect (p>0.05) is simply not evidence for anything, and should be replaced. See Nieuwenhuis, Forstmann and Wagenmakers, (2011) Erroneous Analyses of Interactions in Neuroscience: A Problem of Significance. Nat. Neurosci. 14, 1105-1107, for a more principled discussion of why a lack of significant difference is not the same as a significant interaction. This issue is particularly relevant in this study, that uses small sample sizes as far as statistical power goes, even though I am of course fully aware that larger sample sizes would in so many conditions would not be ethically desirable, but one must be aware that in groups of 8 individuals, and a between group comparison, even large effects (cohen d=0.8) will only be detected in 32% of cases, so even if there is a large effect, you would be more likely not to find it, than to find it, and not finding p<0.05 is thus not evidence for absence even of large effects. After boneferoni correction, things become even worse… This should be carefully addressed throughout the paper, based on ANOVA's or Bayesfactor analyses to qualify statements.

d) Timing of 5HT signals: Line 258+ and Figure 4: it is intriguing that the DeltaF/F seems to increase after onset of the behavior, not prior to onset, if I fully understand the meaning of the '0' point in the Figure 4. The 5HT modulation experiments that you present before seem to suggest causality in the sense of 5HT->social interest and 5HT->allogrooming. For 5HT to trigger those behaviors, one would however expect the fluorescence to precede the behavior, not to follow it. If I understand correctly that the fluorescence follows the behavior rather than preceding them, it would be incompatible with the notion that 5HT triggers the behavior. However 5HT may then maintain the behavior rather than triggering it. Could that be the case? I would suggest providing the following additional data to test this notion: (a) align the fluorescence to the end of the behavior (i.e. moving away from animal in social preference test and stop of grooming or departure from animal in consolation). Would a drop of fluorescence precede the end of the behavior? Second, rather than only plotting the total time in allogrooming or spent with the conspecific, please also present metrics of the duration of the bouts and the number of bouts. If 5HT were to act by prolonging rather than triggering consolation and approach, the bout duration may be a better indicator than the total duration. Please provide this data, and discuss the timing of the fluorescence response and its implications for the role of 5HT in triggering vs. maintaining consolation and approach in the Discussion section.

e) Freezing: Line 482: "if the subject showed evidently immobility, the data were excluded". This statement surprised me, and alerted me to the fact that freezing behavior is not explicitly discussed in the manuscript. Methodologically, I wasn't quite sure what this actually meant: were entire animals removed from the data? How was data-removal performed and how does it affect the presented durations. Does immobility refer to the stressed target of the consolation or the consolator (please specify this throughout the manuscript). Second, freezing has been a key indicator of emotional contagion, and hence of intersubject sensitivity, in the rodent literature. If a stressed demonstrator causes the observer to freeze, this is an important indicator of a reaction to the distress of others, that should really be reported throughout the manuscript. For instance, we know that oxytocin can switch animals between a more active way of coping and a more passive way of coping. If 5HT were to do something like that, it could increase allogrooming, and reduce vicarious freezing in reaction to a stressed demonstrator. Please quantify freezing and immobility, and report it throughout all manipulations, unless it is so rare, that this doesn't make sense. In that case, report briefly how often freezing occurs.

f) Grammar: there are a few grammatical errors in the manuscript. These errors do not compromise the meaning of the manuscript, and are not a big deal, but it may be more elegant to remove them.

*Reviewer #3:*

This manuscript represents a deep-dive into the role of the raphe-to-cingulate serotonergic circuit in empathy-based consoling behavior in the vole. This is a new and underdeveloped literature, and while this is not the first paper to implicate serotonin in empathy-related behavior in rodents, it is certainly the most methodologically thorough. In addition, the use of vole species allows the authors to measure pro-social behavior in a way not possible in other models.

This research is well designed, well executed, well written, and comprehensively answers the question posed. At every stage, the manuscript anticipated my questions and answered them. It will be well-cited in the field. I strongly recommend it for publication.

1. The authors report several behavioral outcome measures in each behavior test. It would be helpful if the authors listed all of the behavioral outcomes measured in the Methods section for each test. Additionally, they should make clear how statistical corrections for multiple comparisons were conducted, if at all. If no corrections were performed, the authors can simply identify in the Methods which behavior was considered the primary outcome measure and which were secondary/exploratory.

*Reviewer #4:*

Overall, this is a straightforward paper in which the authors use orthogonal molecular-genetic approaches to determine the necessity of the DRN→ACC pathway in consolation (e.g. allogrooming of a stressed partner) and social investigation. The paper is impressive given the challenges and limitations of implementing many of these techniques in a non-standard model species, such as the mandarin vole. The paper would be strengthened by applying more rigor with respect to interpretation of results and validation of technical approaches.

1) Throughout the paper the authors refer to empathy-like behaviors, as demonstrated by allogrooming in the consolation test. However, there are marked deficits in general sociability from all their manipulations and no other measures of consolation-specific behaviors, making it likely that effects from inhibition of the DRN→ACC on consolation behaviors are not consolation/empathy-specific, but rather a side effect of disruption of sociability or pro-social behaviors in general. Though this issue is addressed briefly in the discussion, it warrants a more consistent acknowledgment and perhaps a reframing of the paper and the overall interpretation of the results. This does not diminish the importance of the author's findings.

2) The application of the dual-site fiber photometry recording, using a novel 5-HT sensor and GCaMP in a non-standard species is impressive. However, there are a number of concerns regarding the implementation of the approach and the specificity of the 5-HT signal. a) There is no demonstration that the 5-HT sensor is specifically providing a readout for extracellular 5-HT. Given that this sensor has not been extensively used/published in vivo (and never in voles), it is important to show that an antagonist blocks the behavior-associated changes in fluorescence (or another relevant validation approach). b) It is possible that the correlation in dF/F observed in the PFC and in the raphe is not due to 5-HT in the PFC but instead due to GCaMP expressed in DRN terminals in the PFC. This would be ideally addressed by imaging in DRN and PFC of GCaMP-only animals. However, it is worth noting that these concerns are mitigated by the pharmacological results in Figure 5. c) There is no explanation for how the authors controlled for motion artifacts, which are concerning given that their analysis focuses on active behaviors. Despite referring to "multi-channel fibre [sic] photometry" in the methods, they describe recording using only a 480 nm LED, so cannot take advantage of an isosbestic signal. Showing that GFP-expressing animals do not show reliable changes in dF/F during these behaviors would ameliorate these concerns.

[Editors’ note: further revisions were suggested prior to acceptance, as described below.]

Thank you for resubmitting your work entitled "Dorsal raphe nucleus to anterior cingulate cortex 5-HTergic neural circuit modulates consolation and sociability" for further consideration by *eLife*. Your revised article has been evaluated by Kate Wassum (Senior Editor) and a Reviewing Editor.

The manuscript has been improved but there are some remaining issues that need to be addressed, as outlined below:

1) Detail the numbers of animals tested and the number excluded for freezing. It is critical that the reader know the exact number and proportion of animals excluded from each experimental group.

2) The timing of the GCamp signal remains an issue. The timing of which you write: "Interestingly, we found that the fluorescence changes in GCamP6 and 5-HT sensors usually precede allogrooming and social approaching" is not supported by the analysis. See reviewer #3's comments for more detail on this concern.

3) While the title and much of the text along with the results do not distinguish between sociability and consolation, the paragraph on lines 379-388 attempts to make just such a distinction. This should be either changed to bring it in line with the results and remaining text or it should be omitted.

4) Please explain why you would use 8OH-DPAT, an agonist, in the same vein as you used a true 5ht2a antagonist. In the literature, DPAT is often used to reduce 5ht release presumably through effects on presynaptic receptors. But why combine this with the inhibition of 5ht release? Please explain this logic fully. The lines on 339-40 do not explain this sufficiently.

5) In figure 5 E-I it is not clear how the SEM was calculated and it looks as though it is parallel to the average which it would not be if SEM was calculated at each individual time point which is the more appropriate way to convey this information.

6) In the plots to the right of the traces there are only 6 events. It is not clear how many animals were used and which events were used. I think it would be hard to get exactly 6 occurrences of each event.

7) All microscopy images need a scale bar.

8) Please ensure statistical reporting for all main data needs is provided within the main manuscript.

*Reviewer #1:*

This exciting study demonstrates a requirement for serotonin coming from dorsal raphe and terminating in the anterior cingulate cortex for social behavior including approach and allogrooming associated with consolation. Major strengths include the plethora of techniques brought to bear on a focused question. The only substantial quibble I would have with the authors would be in their efforts to distinguish between consolation and sociability (as measured by approach in a 3-chamber test). They show no distinction and thus whether consolation is primary, secondary or parallel to sociability remains entirely unclear.

I still am unclear on how many animals were excluded for freezing. The authors need to articulate how many animals were tested in each group and the number excluded in each group.

I remain less than convinced that the behavior is consolation rather than sociablity. In their rebuttal, the authors compare allogrooming to self-grooming. This is orthogonal to the point. The key point here is that there are changes in the 3-chamber test that accompany each decrease in consolation behavior; the authors interpret these changes as a decrease in sociability. I do not quibble with the data. I do take exception to the efforts to distinguish between sociability and consolation such as the paragraph on lines 379-388. I do not see grounds for any distinction between sociability and consolation, as the title correctly albeit indirectly suggests.

Other concerns have been allayed.

*Reviewer #2:*

This is a re-submission of a manuscript I previously reviewed for this journal. The authors have generally answered all of the questions I posed and made all recommended edits. I have no further comments at this time, though I remain concerned about methodological issues previously raised by other reviewers and satisfactory response to these reviewers' concerns is a necessary precondition for publication.

*Reviewer #3:*

I thank the authors for the thoughtful revision of the paper, and the new data, that further strengthens what is a very strong paper. In particular, the addition of Bayesian statistics, and the careful rephrasing of conclusions in the light of the Bayesian statistics is very helpful. Intuitively, I would have welcomed the inclusion of the key statistics in the figures themselves, rather than in the supplementary 2, but this can be argued either way, and the important factor is that the appropriate statistic is now available and appropriately discussed.

My only remaining comment, which is a nuance rather than a limitation of the paper, regards the photometry signal. In the initial submission, I was surprised that the signal increase followed rather than preceded the consolation behavior. There is now new data, which provides additional evidence that the signal is robustly increased during the social behavior, and together with all the other causal data, this makes a strong case for a relationship between 5HT and these behaviors. The only nuance that I'm still finding difficult to grasp, is the timing relative to onset of the behavior. In the new version you still write: "Interestingly, we found that the fluorescence changes in GCamP6 and 5-HT sensors usually precede allogrooming and social approaching". This is what one would hope for. Unfortunately, I cannot quite see the analysis on which this statement is based. A statistical quantification of the signal in the 100ms before the behavior compared to a further distant baseline might do, or something similar, but currently I really do not see statistical data to support the claim. I would thus suggest either to add a statistical analysis for that, or to remove the statement about this precedence. This is true both in the results and the discussion, where you currently write "Interesting, the increased Ca^2+^ in DR 5-HT neurons often preceded these behaviors which may indicate the motivational function of 5HT (Arakawa, 2020; Yagishita, 2020)". On the other hand, you now provide nice evidence that it decreases before the end of the behavior, in line with the idea that it maintains and motivates the behavior somehow.

*Reviewer #4:*

This is a revised manuscript in which the authors address prior reviewer concerns. Specifically, the authors do a good job of addressing concerns regarding the origin of the fluorescent signal in their fiber photometry experiments, clarifying the logic of the paper, employing Bayesian-based statistical approaches, and other improvements. One remaining concern is regarding the logic underlying the pharmacology experiments. Specifically, figure 6 compares data side by side between 5HT1A agonist and 5HT2A antagonist. It is unclear why it was necessary to block the 5HT2A receptor DURING CNO-mediated inhibition of 5-HT release. If the 5HT2A receptor is important how does blocking it when 5-HT is inhibited by CNO test this?

---

## [Author Response]

[Editors’ note: the authors resubmitted a revised version of the paper for consideration. What follows is the authors’ response to the first round of review.]

This is a multi-pronged study of the contribution of 5ht projections from DR to ACC in consolation behavior in mandarin voles. There was much that the reviewers liked about this study and the manuscript. However, concerns about the photometry data lead us to reject the manuscript, in its present form. There are concerns about the specificity of the method for detecting 5ht. These concerns, well articulated by Reviewer 4 and described below, require new experiments. Should you be able to gather data that assuage these concerns, we would be willing to consider this manuscript as a new submission. While specific reviews are included below, I also highlight here several other concerns that will be critical to address for a successful revision.– Photometry data: We noted that recordings of the 5HT sensor are confounded by the potential presence of GCAMP in ACC terminals. This could be resolved in several ways for example conducting the 5HT and GCAMP recordings in separate subjects. Validation for the selectivity of the 5HT sensor is needed. A control for motion artifact is also needed.

Thank you very much for your good suggestions. We conducted the 5HT sensor and GCAMP recordings in separate animals in the revised version (Figure 4, Figure 5). To validate the selectivity of the 5-HT sensor in mandarin voles, a 5-HT receptor antagonist metergoline (Met) was injected (i.p., 4 mg/kg) before conducting the fiber photometry experiments [1,2]. We found the injection of Met completely blocked the fluorescence change of the 5-HT sensor upon allogrooming (Figure 5—figure supplement 2). In control animals that expressed GFP, we observed little changes in fluorescence which excluded the artifacts of motion (Figure 4, Figure 5).

Citations

1. Onasanwo, S.A., Faborode, S.O., Ilenre, K.O., 2016. Antidepressant-like Potentials of Buchholzia Coriacea Seed Extract: Involvement of Monoaminergic and Cholinergic Systems, and Neuronal Density in the Hippocampus of Adult Mice. Nigerian journal of physiological sciences : official publication of the Physiological Society of Nigeria 31, 93-99.

2. Wan JX, Peng WL, Li XL, Qian TG, Song K, Zeng JZ, et al., A genetically encoded GRAB sensor for measuring serotonin dynamics in vivo. Biorxiv. 2020; 38:569-580; posted February 25, 2020; doi: 10.1101/2020.02.24.962282.

–Is the observed effect attributable specifically to consolation or could it be a more general effect on sociability? The latter possibility should be considered and discussed.

Thank you. As consolation is in general a pro-social behavior, the reduced allogrooming is therefore likely due to the overall decrease in sociability. However, we found both DR 5-HT neurons and ACC 5-HT release showed the highest response during allogrooming (Figure 4—figure supplement 2). Both Burkett’s and our previous study indicated that allogrooming toward a distressed partner is a typical feature of consolation [1,2]. Therefore, we think it is appropriate to say that “DRN→ACC 5-HTergic neural circuit play an important regulatory role in consolation and sociability in mandarin voles”. We discussed this point in the revised version (page 21 line 384-388).

“…As consolation is in general a pro-social behavior, it is difficult to determine whether the reduced allogrooming is due to an overall decrease in sociability. However, in the fiber photometry studies, we found both DR 5-HT neurons and ACC 5-HT release showed the highest responses during allogrooming, which may indicate the specific role of 5-HT in consolation-like behaviors (Figure 4—figure supplement 2.)”

Citations

1. Burkett, J. P., Andari, E., Johnson, Z. V., Curry, D. C., de Waal, F. B., and Young, L. J. (2016). Oxytocin-dependent consolation behavior in rodents. Science, 351(6271), 375-378. doi: 10.1126/science.aac4785

2. Li, L. F., Yuan, W., He, Z. X., Wang, L. M., et al., (2019). Involvement of oxytocin and GABA in consolation behavior elicited by socially defeated individuals in mandarin voles. Psychoneuroendocrinology, 103, 14-24. doi: 10.1016/j.psyneuen.2018.12.238

– The exclusion of data from animals that froze is highly problematic. Please expand on how often this exclusionary criterion was invoked and provide justification for the exclusion.

We are very sorry for the confusion. Actually, on rare occasions in the behavioral test, some tested subjects (not the stimulus voles) show little or no movements, i.e., they just sited there during the test. We think these individuals did not show normal activity and motivation, therefore not appropriate to include their data in the following analysis. The sample sizes are ‘7’ in some cases (Source data file; Figure 1I: Male_ChR2 group, Female_ChR2 group; Figure 1L: Male_Dio group; Figure 2H: Male_Dio group; Figure 3G: Male_GI group; Figure 3H: Female_Dio group, Male_Gq_group, Male_GI group; Figure 3I: Female_Dio group, Male_Gq group, Male_GI_group). In the revised version, we had changed the sentence "…if the subject showed evidently immobility, the data were excluded…" into “…During the test, if the subject (not the stimulus voles) showed very rare or no movements, their data were excluded from the following analysis.…”

– The statistical tests used were not compelling to the reviewers. Reviewer #2 outlines many ways to improve this aspect of the presentation. In addition, the authors should consider straightforward equivalence testing, such as equivalence tests (eg TOST) as well as ANOVAs (and MANOVAs) to address correcting for multiple comparisons (see Reviewer 3).

According to reviewer 2’s suggestion, Bayesian test were conducted for all the analysis and ANOVAs were used for multiple comparisons in the revised version. All the statistical quantifications are presented in Supplementary 2.

– The change in fluorescence follows the behavior of interest. This is not compatible with the causality implied by the authors' interpretation. This needs to be discussed and possibly aspects of the timing re-analyzed along the lines suggested by Reviewer #2.

According to your suggestions, we aligned the GCamp6 and 5-HT signal changes to the end of behaviors and found that DeltaF/F values decreased before the end of allogrooming, social approaching and sniffing (Figure 4—figure supplement 3; Figure 5—figure supplement 1). In consideration of DeltaF/F values reliably increased before these behaviors, we think 5-HT may implicate in some motivational aspect of these behaviors. We discussed this point in the revised version (Page 23, Line 450-452).

– Is consolation behavior at a maximum such that a further enhancement could not be observed with 5ht? The possibility of a ceiling effect as outlined by Reviewer #2 should be considered.

We had discussed this point in the revised version (Page 22, Line 413-417).

“…The other possible explanation is a ceiling effect, which means that the enhancement of consolation and sociability are difficult to accomplish in these normal animals. This may occur in situations where levels of consolation and sociability are low such as in the stressed subjects in our previous study (Li et al., 2020) or genetically less sociable subjects as in Walsh’s study (Walsh et al., 2018)”.

– A schematic showing histological verification of all fiber placements for each experiment is needed. In addition to showing microscopy images showing a larger region of the brain to confirm the image location, it would also help to show a schematic representation of the viral expression spread for each subject (e.g., overlaid on brain slice schematics) for each experiment.

A schematic representation of the viral expression and fiber placements for all the experiments were provided in the revised version (Figure 1—figure supplement 3).

We would be happy to transfer the peer reviews, along with the reviewers' names (where they agree), to another journal of your choice. For example, we have recently facilitated the transfer of review material to a range of journals, including Biology Open, BMC Biology, EMBO Reports, eNeuro, Life Science Alliance, the PLOS journals, the Journal of Cell Biology, Journal of Neuroscience, and the Journal of General Physiology. Such transfers are only undertaken in response to an explicit request by the corresponding author. For neuroscience submissions, please note that we participate within the Neuroscience Peer Review Consortium (https://nprc.incf.org/index.php/about/information-for-authors/).

Reviewer #1:This is a multi-pronged study of the contribution of 5ht projections from DR to ACC in consolation behavior in mandarin voles.There are writing issues that make less than facile to read. However, these issues can be overcome, and the data are extensive and worthy of consideration despite the challenges to the reader.

We appreciate much for your efforts in handling our manuscript. The revised version had been copyedited by AJE (515B8QJY).

The one critical concern that I have is that the authors have not demonstrated that this is consolation behavior as opposed to reunification behavior. In the Burkett et al. study, a very important control of separation without shock was performed. Here that control was not performed. Thus fundamentally the argument that what is being represented here is reunion behavior rather than consolation cannot be refuted by the available evidence.

Actually, the separation without shock was performed as control in our previous study and we found the allogrooming behavior rarely occurred (Figure 1, Figure 3B) [1]. As in Burkett’s study [2], we found grooming towards their shocked partner is an important behavioral indicator of consolation in socially monogamous mandarin voles.

Citations:

1. Burkett, J. P., Andari, E., Johnson, Z. V., Curry, D. C., de Waal, F. B., and Young, L. J. (2016). Oxytocin-dependent consolation behavior in rodents. Science, 351(6271), 375-378. doi: 10.1126/science.aac4785

2. Li, L. F., Yuan, W., He, Z. X., Wang, L. M., et al., (2019). Involvement of oxytocin and GABA in consolation behavior elicited by socially defeated individuals in mandarin voles. Psychoneuroendocrinology, 103, 14-24. doi: 10.1016/j.psyneuen.2018.12.238

The behavior should be described in the Results as well as in the methods. eLife is a journal for the general reader who is unlikely to be familiar with the empathy literature.

Thank you very much for your suggestions. All the behaviors were referred in both the Results and the Method section in the revised version (Results: page 6, line 120-125; Method: page 30, line 552-556).

Are there 5ht terminals in the ACC from median raphe?

It is well known that the serotonergic neurons are mainly located in the dorsal and median raphe nuclei (DR and MnR, respectively) and both sections have bidirectional connections with the prefrontal cortex [1,2]. But in this study, we injected functional DIO virus (ChR2, eNPHR3.0, Gq, Gi) into the DR. Meanwhile, retro-AAVs containing TPH2 promoter and Cre element (rAAV(Retro)-TPH2-Cre) was injected into the ACC. This dual virus strategy ensures that opsins and DREADD are mainly expressed within the DR-ACC 5-HTergic circuits. In spite of this, we think it is interesting and valuable to further investigate the function role of MnR-ACC 5-HTergic circuits in consolation and sociability in future studies.

Citations

1. Fu, W., Le Maitre, E., Fabre, V., Bernard, J.F., David Xu, Z.Q., Hokfelt, T., 2010. Chemical neuroanatomy of the dorsal raphe nucleus and adjacent structures of the mouse brain. The Journal of comparative neurology 518, 3464-3494.

2. Luo, M., Zhou, J., Liu, Z., 2015. Reward processing by the dorsal raphe nucleus: 5-HT and beyond. Learning and memory (Cold Spring Harbor, N.Y.) 22, 452-460.

Make the scales in Figure 4 constant.

Had corrected.

293 Why put a 5ht2ar antagonist while inhibiting DR 5ht neurons? Why not put in an agonist as was done with the 5ht1ar?

We are very sorry for the confusion. It is well-known that 5HT1AR and 5HT2AR are the two main subtypes that expressed at high levels in the mPFC [1,2]. Both in vitro and in vivo evidences indicate that these two types of receptors exert opposite effects in the mPFC. For example, 5-HT1AR coupled to the Gi family of G proteins induces hyperpolarization of pyramidal neurons, whereas 5-HT2AR coupled to Gq proteins induces depolarization in the same neurons [3]. In our previous study, we had indicated that ACC administration of a 5HT1AR agonist (8-OH-DPAT) rescued the reduced consolation and sociability induced by social defeat [4]. We therefore used a 5-HT2AR antagonist here to investigate if this drug could similarly reverse the chemogenetic inhibition effect. We try to make this point more clearly and easier to understand in the revised version (page 18, Line 339-340).

Citations

1. Carhart-Harris, R. L., and Nutt, D. J. (2017). Serotonin and brain function: a tale of two receptors. J Psychopharmacol, 31(9), 1091-1120.

2. Celada, P., Puig, M. V., and Artigas, F. (2013). Serotonin modulation of cortical neurons and networks. Front Integr Neurosci, 7, 25.

3. Carhart-Harris, R.L., Nutt, D.J., 2017. Serotonin and brain function: a tale of two receptors. Journal of psychopharmacology (Oxford, England) 31, 1091-1120.

4. Li, L. F., Yuan, W., He, Z. X., Ma, H., Xun, Y. F., Meng, L. R.,… Tai, F. D. (2020). Reduced consolation behaviors in physically stressed mandarin voles: involvement of oxytocin, dopamine D2 and serotonin 1A receptors within the anterior cingulate cortex. Int J Neuropsychopharmacol. doi: 10.1093/ijnp/pyz060

Reviewer #2:In their paper, Li et al., examine the contribution of serotonin in the ACC to consolation behavior in voles. They show that consolation (as measured as an increase of allogroming towards a mate that has been exposed to shocks), as well as social preference (as measured as more time spent close to a conspecific in a 3 compartment test), are reduced when inhibiting the dorsal raphe->anterior cingulate 5HT pathway, and that this can be rescued using a 5HT1A receptor agonist. In addition, they measure transient increases of 5HT in the ACC following the onset of allogrooming and social approach.In general, this is an interesting and well-conducted study, that combines techniques to interfere with the DR->ACC pathway (including chemo and optogenetic approaches) with some physiology (calcium imaging in DR and 5HT sensors in ACC), to create a series of studies that compel the reader to accept the importance of 5HT on allogroming and social approach.a) Statistic reporting. The paper contains a large number of data, which is great. However, the presentation of the statistical tests and results is often difficult to understand in detail. I applaud the presence of the data-tables for download, and the rigorous illustration of single subject data in most figures. However, results in the text seldom allow to fully understand what statistical test was used. For instance, in the 3 chamber test in Figure 1, the appropriate test would appear to be a group (mcherry vs Chr2) x condition (laser on vs off) interaction in an ANOVA, which is equivalent to a two-sample t-test of the difference On – Off. Please specify in the text and the figures whether an ANOVA looked at this interaction specifically, and always report exact p-values. Also in the figures, when specific t-test were performed, please indicate the test statistic in the figure (e.g. t(n)=x, p=0.0x), whether significant or not, given there is enough space in the figures. In the current version it is often difficult to assess whether statements of effect or no effect are based on ANOVAs (and if so on what interaction or main effect in that ANOVA) or on posthoc t-tests. I know this makes figures a bit more cluttered, but it provides the reader with a richer understanding of the data.

We appreciate much for your good suggestions. In the revised version, all the analysis methods and statistical quantifications are presented in Supplementary 2.

b) Bayesian statistics. The manuscript contains many highly relevant statements that hinge on the fact that a particular condition had no effect on a particular outcome measure, or that a condition brings contagion levels back to normal. These are statements that frequentist statistics (t-test or ANOVA) cannot support, but that are important in the paper. This is because p>0.05 does not provide evidence for H0 (the absence of an effect). Bayes Factor analyses, in contrast, can easily test (see Keysers, Gazzola and Wagenmakers, Nature Neuroscience, 2020 for details of why and how to do that) and quantify evidence for H0. Therefore please always also provide the Bayes factor for the t-tests, and the Bayesfactor for the inclusion of a factor/interaction in an ANOVA. Given that most of your hypotheses are directional, e.g. inhibition of DR->ACC should reduce allogrooming and social preference, I would recommend using directed hypotheses for the Bayesian test to provide more power in favour of H0. This will indicate whether you have evidence that a manipulation had no effect (BF-0<1/3 or BFincl<1/3) or whether you simply have too much variance or two small a group size to establish that a condition had no effect (1/3<BF<3). These analyses can be easily done using JASP on the tables that you provide, and please indicate the BF-0 values together with the t-values in your figures, and adjust your result description according to whether you have evidence of absence or absence of evidence. I know this is some additional work, and people are only starting to adopt BF testing, but this would really help the reader understand when you really have evidence that a manipulation had no effect (BF<1/3) vs. cases in which your group size is simply too small to exclude an effect (see power considerations in point c).

I must applaud these excellent comments and we learn much for that. It is an amazing experience to conduct Bayesian test with JASP. For clear and concise, all the analysis methods and qualifications are presented in Supplementary 2.

c) t-test vs ANOVA: in some cases, your statements call for ANOVA's or Bayesian statistics not t-tests. For instance you write: "Pretreatment with 8-OH-DPAT reversed the dysfunctions to the normal level (all P > 0.05 for '8-OH-DPAT+CNO' vs. 'vehicle+saline' pos-hoc comparisons; Figure 5C-D)". Now, to state that it fully reversed to normal level, you need to show that BF-0 for '8-OH-DPAT+CNO' vs. 'vehicle+saline' is <1/3. If you lack power to establish that, you could use an ANOVA to show that 'Vehicle+Saline – Vehicle+CNO > 8-OH-DPAT+Saline – 8-OH-DPAT+CNO', and say that 8-OH-DPAT+CNO significantly reduces the impact of CNO. Both of these would be evidence for something. Currently, simply showing the lack of a significant effect (p>0.05) is simply not evidence for anything, and should be replaced. See Nieuwenhuis, Forstmann and Wagenmakers, (2011) Erroneous Analyses of Interactions in Neuroscience: A Problem of Significance. Nat. Neurosci. 14, 1105-1107, for a more principled discussion of why a lack of significant difference is not the same as a significant interaction. This issue is particularly relevant in this study, that uses small sample sizes as far as statistical power goes, even though I am of course fully aware that larger sample sizes would in so many conditions would not be ethically desirable, but one must be aware that in groups of 8 individuals, and a between group comparison, even large effects (cohen d=0.8) will only be detected in 32% of cases, so even if there is a large effect, you would be more likely not to find it, than to find it, and not finding p<0.05 is thus not evidence for absence even of large effects. After boneferoni correction, things become even worse… This should be carefully addressed throughout the paper, based on ANOVA's or Bayesfactor analyses to qualify statements.

According to your suggestions, we reanalyzed the data.

Although we can’t find the BF-0 value for '8-OH-DPAT+CNO' vs. 'vehicle+saline' less than 1/3, we did find 'Vehicle+Saline – Vehicle+CNO > 8-OH-DPAT+Saline – 8-OH-DPAT+CNO' (Figure 6_Figure Supplementary 1A(allogrooming): Male: t_(8)_ = 2.375, P = 0.045, BF+0 = 3.750 with median posterior δ = 0.999, 95%CI = [0.091 to 2.493]; Female: t_(10)_ = 2.639, P = 0.025, BF+0 = 5.482 with median posterior δ = 1.075, 95%CI = [0.123 to 2.453]. Figure 6_ Figure Supplementary 1B (social preference ratio): Male: t_(8)_ = 2.727, P = 0.026, BF+0 = 5.368 with median posterior δ = 1.174, 95%CI = [0.123 to 2.767]; Female: t_(10)_ = 2.273, P = 0.046, BF+0 = 3.609 with median posterior δ = 0.906, 95%CI = [0.087 to 2.203]).

We had changed “…Pretreatment with 8-OH-DPAT reversed the dysfunctions to the normal level” into “…Pretreatment with 8-OH-DPAT significantly reduces the impact of CNO…” in the revised version.

d) Timing of 5HT signals: Line 258+ and Figure 4: it is intriguing that the DeltaF/F seems to increase after onset of the behavior, not prior to onset, if I fully understand the meaning of the '0' point in the Figure 4. The 5HT modulation experiments that you present before seem to suggest causality in the sense of 5HT->social interest and 5HT->allogrooming. For 5HT to trigger those behaviors, one would however expect the fluorescence to precede the behavior, not to follow it. If I understand correctly that the fluorescence follows the behavior rather than preceding them, it would be incompatible with the notion that 5HT triggers the behavior. However 5HT may then maintain the behavior rather than triggering it. Could that be the case? I would suggest providing the following additional data to test this notion: (a) align the fluorescence to the end of the behavior (i.e. moving away from animal in social preference test and stop of grooming or departure from animal in consolation). Would a drop of fluorescence precede the end of the behavior? Second, rather than only plotting the total time in allogrooming or spent with the conspecific, please also present metrics of the duration of the bouts and the number of bouts. If 5HT were to act by prolonging rather than triggering consolation and approach, the bout duration may be a better indicator than the total duration. Please provide this data, and discuss the timing of the fluorescence response and its implications for the role of 5HT in triggering vs. maintaining consolation and approach in the Discussion section.

As suggested, we replicated the GCamp6 and 5-HT fiber photometric experiments in separated animals and found that both the signals reliably increased before allogrooming, social approaching and sniffing (Figure 4 and 5). When aligning the fluorescence signals to the end of these behaviors, the DeltaF/F values invariably decreased before withdrawing from these behaviors (Figure 4—figure supplement 3A-3B; Figure 5—figure supplement 1). All the results indicate that 5HT may be involved in some motivational aspect of these behaviors. This point had been discussed in the revised version (page 14, line 258-259; page 24, line 451-452).

We reviewed our data and found the inhibition manipulations largely impact the number of bouts rather than the bout duration of allogrooming and chasing, etc. This is consistent with the motivational effect of 5-HT. For concise and clear presentation, these data are not presented in the manuscript, but we presented one group of data affiliated with Figure 1H for your reference.

e) Freezing: Line 482: "if the subject showed evidently immobility, the data were excluded". This statement surprised me, and alerted me to the fact that freezing behavior is not explicitly discussed in the manuscript. Methodologically, I wasn't quite sure what this actually meant: were entire animals removed from the data? How was data-removal performed and how does it affect the presented durations. Does immobility refer to the stressed target of the consolation or the consolator (please specify this throughout the manuscript). Second, freezing has been a key indicator of emotional contagion, and hence of intersubject sensitivity, in the rodent literature. If a stressed demonstrator causes the observer to freeze, this is an important indicator of a reaction to the distress of others, that should really be reported throughout the manuscript. For instance, we know that oxytocin can switch animals between a more active way of coping and a more passive way of coping. If 5HT were to do something like that, it could increase allogrooming, and reduce vicarious freezing in reaction to a stressed demonstrator. Please quantify freezing and immobility, and report it throughout all manipulations, unless it is so rare, that this doesn't make sense. In that case, report briefly how often freezing occurs.

We are very sorry for the confusion. Actually, on rare occasions in the behavioral test, some tested subjects (not the stimulus voles) show little or no movements, i.e., they just sited there during all the testing period. We then think it is not appropriate to include these statistics in the following analysis. The sample size is then ‘7’ in some cases (Source data file; Figure 1I: Male_ChR2 group, Female_ChR2 group; Figure 1L: Male_Dio group; Figure 2H: Male_Dio group; Figure 3G: Male_GI group; Figure 3H: Female_Dio group, Male_Gq_group, Male_GI group; Figure 3I: Female_Dio group, Male_Gq group, Male_GI_group). In the revised version, we had changed the sentence "…if the subject showed evidently immobility, the data were excluded…" into “…if the tested subject (not the stimulus voles) showed very little or no movements during the test, the data were excluded from the following analysis…”

For another, we agree with your opinion that freezing is an important emotional indicator in rodents, but it is seldom occurred in our experiments. A paired conditioned stimulus may elicit such response, which just as in Burkett’s study had done. However, we analyzed self-grooming behavior in our experiments which is another important emotional indicator in rodents (Figure 1—figure supplement 4; Figure 2—figure supplement 1; Figure 3—figure supplement 2; Figure 4; Figure 5 and Figure 6—figure supplement 2).

Citations

Burkett, J. P., Andari, E., Johnson, Z. V., Curry, D. C., de Waal, F. B., and Young, L. J. (2016). Oxytocin-dependent consolation behavior in rodents. Science, 351(6271), 375-378. doi: 10.1126/science.aac4785

f) Grammar: there are a few grammatical errors in the manuscript. These errors do not compromise the meaning of the manuscript, and are not a big deal, but it may be more elegant to remove them.

The revised version had been copyedited by AJE (515B8QJY).

Reviewer #3:This manuscript represents a deep-dive into the role of the raphe-to-cingulate serotonergic circuit in empathy-based consoling behavior in the vole. This is a new and underdeveloped literature, and while this is not the first paper to implicate serotonin in empathy-related behavior in rodents, it is certainly the most methodologically thorough. In addition, the use of vole species allows the authors to measure pro-social behavior in a way not possible in other models.This research is well designed, well executed, well written, and comprehensively answers the question posed. At every stage, the manuscript anticipated my questions and answered them. It will be well-cited in the field. I strongly recommend it for publication.

We appreciated much for your comments to our manuscript.

1. The authors report several behavioral outcome measures in each behavior test. It would be helpful if the authors listed all of the behavioral outcomes measured in the Methods section for each test. Additionally, they should make clear how statistical corrections for multiple comparisons were conducted, if at all. If no corrections were performed, the authors can simply identify in the Methods which behavior was considered the primary outcome measure and which were secondary/exploratory.

All the analyzed behaviors were referred in both the Results and the Method section in the revised version (Results: page 6, line 122-124; Method: page 30, line 553-556). According to reviewer 2’s suggestions, Bayesian test were conducted for all the analysis and ANOVAs were used for multiple comparisons in the revised version. All analysis methods and the statistical quantifications are presented in Supplementary 2.

Reviewer #4:Overall, this is a straightforward paper in which the authors use orthogonal molecular-genetic approaches to determine the necessity of the DRN→ACC pathway in consolation (e.g. allogrooming of a stressed partner) and social investigation. The paper is impressive given the challenges and limitations of implementing many of these techniques in a non-standard model species, such as the mandarin vole. The paper would be strengthened by applying more rigor with respect to interpretation of results and validation of technical approaches.1) Throughout the paper the authors refer to empathy-like behaviors, as demonstrated by allogrooming in the consolation test. However, there are marked deficits in general sociability from all their manipulations and no other measures of consolation-specific behaviors, making it likely that effects from inhibition of the DRN→ACC on consolation behaviors are not consolation/empathy-specific, but rather a side effect of disruption of sociability or pro-social behaviors in general. Though this issue is addressed briefly in the discussion, it warrants a more consistent acknowledgment and perhaps a reframing of the paper and the overall interpretation of the results. This does not diminish the importance of the author's findings.

Thank you very much for your suggestions. Yes, we agree with your viewpoint that the reduced consolation induced by inhibition of the DRN→ACC is likely due to the overall decrease in sociability. However, in the fiber photometry studies, we found both DR 5-HT neurons and ACC 5-HT release showed the largest response during allogrooming (Figure 4—figure supplement 2). Both Burkett’s and our previous study indicated that allogrooming toward a distressed partner is a typical feature of consolation [1,2]. Therefore, we think it is appropriate to say that “DRN→ACC 5-HTergic neural circuit play an important regulatory role in consolation and sociability in mandarin voles”. We discussed this point in the revised version (page 21 line 384-388).

Citations

1. Burkett, J. P., Andari, E., Johnson, Z. V., Curry, D. C., de Waal, F. B., and Young, L. J., (2016). Oxytocin-dependent consolation behavior in rodents. Science, 351(6271), 375-378. doi: 10.1126/science.aac4785

2. Li, L. F., Yuan, W., He, Z. X., Wang, L. M., et al., (2019). Involvement of oxytocin and GABA in consolation behavior elicited by socially defeated individuals in mandarin voles. Psychoneuroendocrinology, 103, 14-24. doi: 10.1016/j.psyneuen.2018.12.238

2) The application of the dual-site fiber photometry recording, using a novel 5-HT sensor and GCaMP in a non-standard species is impressive. However, there are a number of concerns regarding the implementation of the approach and the specificity of the 5-HT signal. a) There is no demonstration that the 5-HT sensor is specifically providing a readout for extracellular 5-HT. Given that this sensor has not been extensively used/published in vivo (and never in voles), it is important to show that an antagonist blocks the behavior-associated changes in fluorescence (or another relevant validation approach). b) It is possible that the correlation in dF/F observed in the PFC and in the raphe is not due to 5-HT in the PFC but instead due to GCaMP expressed in DRN terminals in the PFC. This would be ideally addressed by imaging in DRN and PFC of GCaMP-only animals. However, it is worth noting that these concerns are mitigated by the pharmacological results in Figure 5. c) There is no explanation for how the authors controlled for motion artifacts, which are concerning given that their analysis focuses on active behaviors. Despite referring to "multi-channel fibre [sic] photometry" in the methods, they describe recording using only a 480 nm LED, so cannot take advantage of an isosbestic signal. Showing that GFP-expressing animals do not show reliable changes in dF/F during these behaviors would ameliorate these concerns.

We appreciate much for these excellent comments！According to your suggestions, we conducted the 5HT sensor and GCAMP6 recordings in separate animals in the revised version (Figure 4, Figure 5). Meanwhile, to validate the selectivity of this 5-HT sensor in mandarin voles, a 5-HT receptor antagonist metergoline (Met) was injected (i.p., 4 mg/kg) before conducting fiber photometry recordings. The results showed that Met completely blocked the fluorescence change of the 5-HT sensor along with allogrooming (Figure 5 figure supplement 2). Finally, control animals only expressing GFP were examined in additional experiments (Figure 4, 5), and we found little fluorescence changes along with all the behaviors which excluding the motion

[Editors’ note: what follows is the authors’ response to the second round of review.]

The manuscript has been improved but there are some remaining issues that need to be addressed, as outlined below:1) Detail the numbers of animals tested and the number excluded for freezing. It is critical that the reader know the exact number and proportion of animals excluded from each experimental group.

Both the animal numbers and the animals excluded from analysis due to immobility (Figure 1I, Figure 1L and Figure 2H) are listed in the revised figure legends.

2) The timing of the GCamp signal remains an issue. The timing of which you write: "Interestingly, we found that the fluorescence changes in GCamP6 and 5-HT sensors usually precede allogrooming and social approaching" is not supported by the analysis. See reviewer #3's comments for more detail on this concern.

We deleted this statement in the revised version.

3) While the title and much of the text along with the results do not distinguish between sociability and consolation, the paragraph on lines 379-388 attempts to make just such a distinction. This should be either changed to bring it in line with the results and remaining text or it should be omitted.

Admittedly, it is difficult to clarify whether the decrease in sociability is the reason behind the reduced consolation in our study. We therefore try not to distinguish the two in the revised version and just say “…As consolation is in general a pro-social behavior, the reduced allogrooming may be largely due to an overall decrease in sociability. Further experiments are needed to address this interesting question” (page 12, line 279-281).

4) Please explain why you would use 8OH-DPAT, an agonist, in the same vein as you used a true 5ht2a antagonist. In the literature, DPAT is often used to reduce 5ht release presumably through effects on presynaptic receptors. But why combine this with the inhibition of 5ht release? Please explain this logic fully. The lines on 339-40 do not explain this sufficiently.

Actually, 5-HT1A receptors are found in two main populations of neurons: (1) serotonin neurons of the midbrain raphe nuclei (as presynaptic autoreceptors to control the release of 5-HT); and (2) postsynaptic neurons mainly in the hippocampus, septum, amygdala and corticolimbic areas as to modulate neuronal excitability and plasticity.^[1]^ The microinjection of 8OH-DPAT into the ACC may mainly interact with postsynaptic receptors there.

Both 5-HT1AR and 5-HT2AR express abundantly in the mPFC.^[2]^ In our previous study, we found that ACC administration of 8-OH-DPAT rescued the impaired consolation and sociability induced by social defeat.^[3]^ We then used 8-OH-DPAT to investigate whether it could similarly relieve the impaired consolation and sociability induced by inhibition of DR-ACC 5-HTergic circuit, which aimed to clarify the involvement of 5-HT1AR during this process.

Increasing evidences indicate that 5-HT1AR and 5-HT2AR exert opposite effects. For example, 5-HT1AR coupled to the Gi family of G proteins induces hyperpolarization of pyramidal neurons, whereas 5-HT2AR coupled to Gq proteins induces depolarization in the same neurons.^[4]^ Preclinical studies had indicated that 5-HT2A antagonists exerted anxiolytic effects.^[5]^ Therefore, we used MDL 100907 (a 5-HT2A antagonist) here to investigate the possible involvement of 5-HT2AR during this process. We rephrase the sentence in the revised version to make it more logical in the revised version (Page 11, line 246-255).

“…The most abundant 5-HT receptors expressed in the mPFC are 5-HT1AR and 5-HT2AR [4]. In our previous study, we indicated that ACC administration of a 5HT1AR agonist (8-OH-DPAT) rescued the impaired consolation and sociability induced by chronic social defeat [3]. In the same vein, we used this drug to investigate whether it could relieve the impaired consolation and sociability induced by the inhibition of DR-ACC 5-HTergic circuit, which aimed to clarify the involvement of 5-HT1AR during this process. Considering 5-HT1AR and 5-HT2AR usually exert opposite effects within the mPFC [2, 6], and 5-HT2A antagonists had been reported to exert anxiolytic effects in preclinical studies, we then used MDL 100907 (a 5-HT2A antagonist) to investigate the possible involvement of 5-HT2AR during this process…”

Citations

1. Artigas, F., Serotonin receptors involved in antidepressant effects. Pharmacol Ther, 2013. 137(1): p. 119-31.

2. Celada, P., M.V. Puig, and F. Artigas, Serotonin modulation of cortical neurons and networks. Front Integr Neurosci, 2013. 7: p. 25.

3. Li, L.F., et al., Reduced Consolation Behaviors in Physically Stressed Mandarin Voles: Involvement of Oxytocin, Dopamine D2, and Serotonin 1A Receptors Within the Anterior Cingulate Cortex. Int J Neuropsychopharmacol, 2020. 23(8): p. 511-523.

4. Carhart-Harris, R.L. and D.J. Nutt, Serotonin and brain function: a tale of two receptors. J Psychopharmacol, 2017. 31(9): p. 1091-1120.

5. Carson, W.H. and H. Kitagawa, Drug development for anxiety disorders: new roles for atypical antipsychotics. Psychopharmacol Bull, 2004. 38(1): p. 38-45.

5. Puig, M.V. and A.T. Gulledge, Serotonin and prefrontal cortex function: neurons, networks, and circuits. Mol Neurobiol, 2011. 44(3): p. 449-64.

5) In figure 5 E-I it is not clear how the SEM was calculated and it looks as though it is parallel to the average which it would not be if SEM was calculated at each individual time point which is the more appropriate way to convey this information.

The representative images of fluorescence signals in Figure 4 and 5 are exported from a custom-written MATLAB software package developed by ThinkerTech (Nanjing, China). The original source code file could be found in GitHub,

(https://github.com/ganenlife1980/MultiFiberPhotometry__MetLab).

We asked an engineer from this company and he replied that the shades in the images are ‘SEM’ of the signals which reflecting the range and trend of variations. Similar presentations could be found in Wei’s (Figure 1d),^[1]^ Seo’s B (Figure 2e), ^[2]^ Yuan’s (Figure 1G),^[3]^ Tang’s (Figure 2I, J)^[4]^ and Flanigan’s^[5]^ (Figures 1 and 3) paper.

Citations:

1. Wei, Y.C., et al., Medial preoptic area in mice is capable of mediating sexually dimorphic behaviors regardless of gender. Nat Commun, 2018. 9(1): p. 279.

2. Seo, C., et al., Intense threat switches dorsal raphe serotonin neurons to a paradoxical operational mode. Science, 2019. 363(6426): p. 538-542.

3. Yuan, Y., et al., Reward Inhibits Paraventricular CRH Neurons to Relieve Stress. Curr Biol, 2019. 29(7): p. 1243-1251.e4.

4. Tang, Y., et al., Social touch promotes interfemale communication via activation of parvocellular oxytocin neurons. Nat Neurosci, 2020. 23(9): p. 1125-1137.

5. Flanigan, M.E., et al., Orexin signaling in GABAergic lateral habenula neurons modulates aggressive behavior in male mice. Nat Neurosci, 2020. 23(5): p. 638-650.

6) In the plots to the right of the traces there are only 6 events. It is not clear how many animals were used and which events were used. I think it would be hard to get exactly 6 occurrences of each event.

We are sorry for the confusion. During the test, the signals of 4-6 bouts of a specific behavior of an animal were collected. The averaged bouts data are then served as the representative value for this behavior of an individual. The details and animal numbers are indicated in the figure legends of the revised version.

7) All microscopy images need a scale bar.

Had added as request.

8) Please ensure statistical reporting for all main data needs is provided within the main manuscript.

All the statistical quantifications are presented in the figure legends in the revised version.

Reviewer #1:This exciting study demonstrates a requirement for serotonin coming from dorsal raphe and terminating in the anterior cingulate cortex for social behavior including approach and allogrooming associated with consolation. Major strengths include the plethora of techniques brought to bear on a focused question. The only substantial quibble I would have with the authors would be in their efforts to distinguish between consolation and sociability (as measured by approach in a 3-chamber test). They show no distinction and thus whether consolation is primary, secondary or parallel to sociability remains entirely unclear.

Admittedly, it is difficult to clarify whether the decrease in sociability is the reasons behind the reduced consolation in our study. We try not to distinguish the two in the revised version and just say “…As consolation is in general a pro-social behavior, the reduced allogrooming may be largely due to an overall decrease in sociability. Further experiments are needed to address this interesting question” (page 12, line 279-281).

I still am unclear on how many animals were excluded for freezing. The authors need to articulate how many animals were tested in each group and the number excluded in each group.

Both the animal numbers and the animals excluded from analysis (Figure 1I, Figure 1L and Figure 2H) are listed in the revised figure legends.

I remain less than convinced that the behavior is consolation rather than sociablity. In their rebuttal, the authors compare allogrooming to self-grooming. This is orthogonal to the point. The key point here is that there are changes in the 3-chamber test that accompany each decrease in consolation behavior; the authors interpret these changes as a decrease in sociability. I do not quibble with the data. I do take exception to the efforts to distinguish between sociability and consolation such as the paragraph on lines 379-388. I do not see grounds for any distinction between sociability and consolation, as the title correctly albeit indirectly suggests.

Admittedly, it is difficult to clarify whether the decrease in sociability is the reasons behind the reduced consolation in our study. We try not to distinguish the two in the revised version and just say “…As consolation is in general a pro-social behavior, the reduced allogrooming here may be largely due to an overall decrease in sociability. Further experiments are needed to address this interesting question” (page 12, line 279-281).

Reviewer #3:I thank the authors for the thoughtful revision of the paper, and the new data, that further strengthens what is a very strong paper. In particular, the addition of Bayesian statistics, and the careful rephrasing of conclusions in the light of the Bayesian statistics is very helpful. Intuitively, I would have welcome the inclusion of the key statistics in the figures themselves, rather than in the supplementary 2, but this can be argued either way, and the important factor is that the appropriate statistic is now available and appropriately discussed.My only remaining comment, which is a nuance rather than a limitation of the paper, regards the photometry signal. In the initial submission, I was surprised that the signal increase followed rather than preceded the consolation behavior. There is now new data, which provides additional evidence that the signal is robustly increased during the social behavior, and together with all the other causal data, this makes a strong case for a relationship between 5HT and these behaviors. The only nuance that I'm still finding difficult to grasp, is the timing relative to onset of the behavior. In the new version you still write: "Interestingly, we found that the fluorescence changes in GCamP6 and 5-HT sensors usually precede allogrooming and social approaching". This is what one would hope for. Unfortunately, I cannot quite see the analysis on which this statement is based. A statistical quantification of the signal in the 100ms before the behavior compared to a further distant baseline might do, or something similar, but currently I really do not see statistical data to support the claim. I would thus suggest either to add a statistical analysis for that, or to remove the statement about this precedence. This is true both in the results and the discussion, where you currently write "Interesting, the increased Ca^2+^ in DR 5-HT neurons often preceded these behaviors which may indicate the motivational function of 5HT (Arakawa, 2020; Yagishita, 2020)". On the other hand, you now provide nice evidence that it decreases before the end of the behavior, in line with the idea that it maintains and motivates the behavior somehow.

Firstly, we appreciate much for your efforts in helping improve our manuscript. In the revised version, we removed this statement.

Reviewer #4:This is a revised manuscript in which the authors address prior reviewer concerns. Specifically, the authors do a good job of addressing concerns regarding the origin of the fluorescent signal in their fiber photometry experiments, clarifying the logic of the paper, employing Bayesian-based statistical approaches, and other improvements. One remaining concern is regarding the logic underlying the pharmacology experiments. Specifically, figure 6 compares data side by side between 5HT1A agonist and 5HT2A antagonist. It is unclear why it was necessary to block the 5HT2A receptor DURING CNO-mediated inhibition of 5-HT release. If the 5HT2A receptor is important how does blocking it when 5-HT is inhibited by CNO test this?

It is well-known that 5-HT1AR and 5-HT2AR express abundantly in the mPFC.^[1]^ In our previous study, we found that administration of 8-OH-DPAT in the ACC rescued the impaired consolation and sociability induced by social defeat.^[2]^ We then used 8-OH-DPAT here to investigate whether it could similarly relieve the impaired consolation and sociability induced by inhibition of DR-ACC 5-HTergic circuit, which aimed to clarify the involvement of 5-HT1AR during this process.

Increasing evidences indicate that 5-HT1AR and 5-HT2AR exert opposite effects. For example, 5-HT1AR coupled to the Gi family of G proteins induces hyperpolarization of pyramidal neurons, whereas 5-HT2AR coupled to Gq proteins induces depolarization in the same neurons.^[3]^ Preclinical studies had indicated that 5-HT2A antagonists exerted anxiolytic effects.^[4]^ Therefore, we used MDL 100907 (a 5-HT2A antagonist) here to investigate the involvement of 5-HT2AR during this process. We rephrase the sentence in the revised version to make it more easier to understand in the revised version (Page 11, line 246-255).

“…The most abundant 5-HT receptors expressed in the mPFC are 5-HT1AR and 5-HT2AR [3]. In our previous study, we indicated that ACC administration of a 5HT1AR agonist (8-OH-DPAT) rescued the impaired consolation and sociability induced by chronic social defeat [2]. In the same vein, we used this drug to investigate whether it could relieve the impaired consolation and sociability induced by the inhibition of DR-ACC 5-HTergic circuit, which aiming to clarify the involvement of 5-HT1AR during this process. Considering 5-HT1AR and 5-HT2AR usually exert opposite effects within the mPFC [1, 5], and 5-HT2A antagonists had been reported to exert anxiolytic effects in preclinical studies, we then used MDL 100907 (a 5-HT2A antagonist) to investigate the possible involvement of 5-HT2AR during this process…”

Citations

1. Celada, P., M.V. Puig, and F. Artigas, Serotonin modulation of cortical neurons and networks. Front Integr Neurosci, 2013. 7: p. 25.

2. Li, L.F., et al., Reduced Consolation Behaviors in Physically Stressed Mandarin Voles: Involvement of Oxytocin, Dopamine D2, and Serotonin 1A Receptors Within the Anterior Cingulate Cortex. Int J Neuropsychopharmacol, 2020. 23(8): p. 511-523.

3. Carhart-Harris, R.L. and D.J. Nutt, Serotonin and brain function: a tale of two receptors. J Psychopharmacol, 2017. 31(9): p. 1091-1120.

4. Carson, W.H. and H. Kitagawa, Drug development for anxiety disorders: new roles for atypical antipsychotics. Psychopharmacol Bull, 2004. 38(1): p. 38-45.

5. Puig, M.V. and A.T. Gulledge, Serotonin and prefrontal cortex function: neurons, networks, and circuits. Mol Neurobiol, 2011. 44(3): p. 449-64.